# ON THE CONVERGENCE OF FEDPROX WITH EXTRAPOLATION AND INEXACT PROX

## ABSTRACT

Enhancing the FedProx federated learning algorithm (Li et al., 2020) with server-side extrapolation, Li et al. (2024a) recently introduced the FedExProx method. Their theoretical analysis, however, relies on the assumption that each client computes a certain proximal operator exactly, which is impractical since this is virtually never possible to do in real settings. In this paper, we investigate the behavior of FedExProx without this exactness assumption in the smooth and globally strongly convex setting. We establish a general convergence result, showing that inexactness leads to convergence to a neighborhood of the solution. Additionally, we demonstrate that, with careful control, the adverse effects of this inexactness can be mitigated. By linking inexactness to biased compression (Beznosikov et al., 2023), we refine our analysis, highlighting robustness of extrapolation to inexact proximal updates. We also examine the local iteration complexity required by each client to achieved the required level of inexactness using various local optimizers. Our theoretical insights are validated through comprehensive numerical experiments.

## 1 INTRODUCTION

Distributed optimization is becoming increasingly essential in modern machine learning, especially as models grow more complex. Federated learning (FL), a decentralized approach where multiple clients collaboratively train a shared model while keeping their data locally to preserve privacy, is a key example of this trend (Konečný et al., 2016; McMahan et al., 2017). Often, a central server coordinates the process by aggregating the locally trained models from each client to update the global model without accessing the raw data. The federated average algorithm (FedAvg), introduced by McMahan et al. (2017) and Mangasarian & Solodov (1993), is one of the most popular strategies for tackling federated learning problems. The algorithm comprises three essential components: client sampling, data sampling, and local training. During its execution, the server first samples a subset of clients to participate in the training process for a given round. Each selected client then performs local training using stochastic gradient descent (SGD), with or without random reshuffling, to enhance communication efficiency, as documented by Bubeck et al. (2015); Gower et al. (2019); Moulines & Bach (2011); Sadiev et al. (2022b). FedAvg has proven to be highly successful in practice, nevertheless it suffers from client drift when data is heterogeneous (Karimireddy et al., 2020).

Various techniques have been proposed to address the challenges of data heterogeneity, with FedProx, introduced by Li et al. (2020), being one notable example. Rather than having each client perform local SGD rounds, FedProx requires each client to compute a proximal operator locally. Computing the proximal operator can be regarded as an optimization problem that each client can solve locally. Proximal algorithms are advantageous when the proximal operators can be evaluated relatively easily (Parikh et al., 2014). Algorithms based on proximal operators, such as the proximal point method (PPM) (Rockafellar, 1976; Parikh et al., 2014) and its extension to the stochastic setting (SPPM) (Bertsekas, 2011; Asi & Duchi, 2019; Khaled & Jin, 2022; Richtárik & Takáč, 2020; Patrascu & Necoara, 2018), offer greater stability against inaccurately specified step sizes, unlike gradient-based methods. PPM was introduced by Martinet (1972) and expanded by Rockafellar (1976). Its extension into the stochastic setting are often used in federated optimization. The stability mentioned is particularly useful when problem-specific parameters, such as the smoothness constant of the objective function, are unknown which renders determining the step size for SGD

becomes challenging. Indeed, an excessively large step size in SGD leads to divergence, while a small step size ensures convergence but significantly slows down the training process.

Another approach to mitigating the slowdown caused by heterogeneity is the use of a server step size. Specifically, in FedAvg, a local step size is employed by each client to minimize their individual objectives, while a server step size is used to aggregate the 'pseudo-gradients' obtained from each client (Karimireddy et al., 2020; Reddi et al., 2021). The local step size is set relatively small to mitigate client drift, while the server step size is set larger to avoid slowdowns. However, the small step sizes result in a slowdown during the initial phase of training, which cannot be fully compensated by the large server step size (Jhunjhunwala et al., 2023). Building on the extrapolation technique employed in parallel projection methods to solve the convex feasibility problem (Censor et al., 2001; Combettes, 1997; Necoara et al., 2019), Jhunjhunwala et al. (2023) introduced FedExP as an extension of FedAvg, incorporating adaptive extrapolation as the server step size. Extrapolation involves moving further along the line connecting the most recent iterate, $x_k$, and the average of the projections of $x_k$ onto different convex sets, $\mathcal{X}_i$, in the parallel projection method, which accelerates the algorithm. Extrapolation is also known as over-relaxation (Rechardson, 1911) in fixed point theory. It is a common technique to effectively accelerate the convergence of fixed point methods including gradient based algorithms and proximal splitting algorithms (Condat et al., 2023; Iutzeler & Hendrickx, 2019). Recently, Li et al. (2024a) shows that the combination of extrapolation with FedProx also results in better complexity bounds. The analysis of the resulting algorithm FedExProx reveals the relationship between the extrapolation parameter and the step size of gradient-based methods with respect to the Moreau envelope associated with the original objective function.[1] However, it relies on the assumption that each proximal operator is solved accurately, which makes it impractical and less advantageous compared to gradient-based algorithms.

## 1.1 Contributions

Our paper makes the following contributions, please refer to Appendix A for notation details.

- We provide a new analysis of FedExProx based on Li et al. (2024a), focusing on the case where the proximal operators are evaluated inexactly in the globally strongly convex setting, removing the need for the assumption of exact proximal operator evaluations. By properly defining the notion of approximation, we establish a general convergence guarantee of the algorithm to a neighborhood of the solution utilizing the theory of biased SGD (Demidovich et al., 2024). Specifically, our algorithm achieves a linear convergence rate of $\mathcal{O}\left(\frac{L_\gamma(1+\gamma L_{\max})}{\mu}\right)$ to a neighborhood of the solution, matching the rate presented by Li et al. (2024a).
- Building on our understanding of how the neighborhood arises, we propose a new method of approximation. This alternative characterization of inexactness eliminates the neighborhood from the previous convergence guarantee, provided that the inexactness is properly bounded, and the extrapolation parameter is chosen to be sufficiently small.
- By leveraging the similarity between the definitions of inexactness and compression, we enhance our analysis using the theory of biased compression (Beznosikov et al., 2023). The improved analysis offers a faster rate of $\mathcal{O}\left(\frac{L_\gamma(1+\gamma L_{\max})}{\mu-4\varepsilon_2 L_{\max}}\right)^2$, leading to convergence to the exact solution, provided that the inexactness is bounded in a more permissive manner. More importantly, the optimal extrapolation $1/\gamma L_\gamma$ matches the exact case. This shows that extrapolation aids convergence as long as sufficient accuracy is reached, even with inexact proximal evaluations.
- We then analyze how the aforementioned approximations can be obtained by each client. As examples, we provide the local iteration complexity when the client employs gradient descent (GD) or Nesterov's accelerated gradient descent (AGD), demonstrating that these approximations are readily achievable. Specifically, for the $i$-th client, the local iteration complexity is $\tilde{\mathcal{O}}\left(1+\gamma L_i\right)$ when using GD, and $\tilde{\mathcal{O}}\left(\sqrt{1+\gamma L_i}\right)$ when using AGD. See Table 1 and Table 2 for a detailed comparison of complexities of all relevant quantities.

---

[1]A tighter convergence guarantee in some cases is obtained by Anyszka et al. (2024).

[2]The parameter $\varepsilon_2$ is the parameter associated with accuracy of relative approximation as defined in Definition 4. We use the notation $\mathcal{O}\left(\cdot\right)$ to ignore constant factors and $\tilde{\mathcal{O}}\left(\cdot\right)$ when logarithmic factors are also omitted.

Table 1: Comparison of FedExProx (Li et al., 2024a) and our proposed inexact versions of the algorithms using different approximations. In the convergence column, we present the rate at which each algorithm converges to either the solution or a neighborhood in the globally strongly convex setting. Here, $L_\gamma$ represents the smoothness constant of $M^\gamma$ as defined before Theorem 1. The neighborhood column indicates the size of the neighborhood, while the optimal extrapolation column suggests the best choice of $\alpha$ for each algorithm. The final column outlines the conditions on the inexactness. All quantities are presented with constant factors omitted, $K$ is the number of total iterations, $\gamma$ is the local step size for the proximal operator, $S(\varepsilon_2)$ defined in Theorem 2 is a factor of slowing down due to inexactness in $(0, 1]$. For relative approximation, we first present the original theory in the third row and then place the sharper analysis in the following row for comparison.

| Algorithm | Convergence | Neighborhood | Optimal Extrapolation | Bound on Inexactness |
|---|---|---|---|---|
| FedExProx | $\exp\left(-\frac{K\mu}{L_\gamma(1+\gamma L_{\max})}\right)$ | $0$ | $\frac{1}{\gamma L_\gamma}$ | NA |
| (NEW) FedExProx with $\varepsilon_1$ approximation | $\exp\left(-\frac{K\mu}{L_\gamma(1+\gamma L_{\max})}\right)$ | $\varepsilon_1\left(\frac{\frac{1}{\gamma}+L_{\max}}{\mu}\right)^2$ [a] | $\frac{1}{4\gamma L_\gamma}$ [b] | NA [c] |
| (NEW) FedExProx with $\varepsilon_2$ relative approximation by biased SGD | $\exp\left(-\frac{K\mu S(\varepsilon_2)}{L_\gamma(1+\gamma L_{\max})}\right)$ [d] | $0$ | $< \frac{1}{\gamma L_\gamma}$ | $< \frac{\mu^2}{4L_{\max}^2}$ |
| (NEW) FedExProx with $\varepsilon_2$ relative approximation by biased compression | $\exp\left(-\frac{K(\mu-4\varepsilon_2 L_{\max})}{L_\gamma(1+\gamma L_{\max})}\right)$ | $0$ | $\frac{1}{\gamma L_\gamma}$ [e] | $< \frac{\mu}{4L_{\max}}$ |

[a] Note that when $\varepsilon_1 = 0$, i.e., when the proximal operators are evaluated exactly, the neighborhood diminishes, and we recover the result of FedExProx by Li et al. (2024a), up to a constant factor.

[b] The optimal extrapolation parameter here is 4 times smaller than the exact case, results in a slightly slower convergence. Note that constant factors for convergence are ommited in the table.

[c] Unlike relative approximations, the convergence guarantee here is more general, allowing for the analysis of unbounded inexactness. However, as the inexactness increases, the neighborhood grows correspondingly, rendering the result practically useless.

[d] Refer to Theorem 2 for the definition of $S(\varepsilon_2)$ and the corresponding optimal extrapolation parameter. The theory indicates that inexactness will adversely affect the algorithm's convergence.

[e] Surprisingly, our sharper analysis reveals that the optimal extrapolation parameter in this case remains the same as in the exact setting, highlighting the effectiveness of extrapolation even when the proximal operators are evaluated inexactly.

Table 2: Comparison of local iteration complexities of each client in order to obtain an approximation using either GD or AGD (Nesterov, 2004). We use the $i$-th client as an example, where the local objective $f_i : \mathbb{R}^d \mapsto \mathbb{R}$ is $L_i$-smooth and convex, $i \in \{1, 2, \ldots, n\}$.

| Algorithm | $\varepsilon_1$ absolute approximation | $\varepsilon_2$ relative approximation |
|---|---|---|
| Gradient descent | $\mathcal{O}\left((1+\gamma L_i)\log\left(\frac{\|x_k-\text{prox}_{\gamma f_i}(x_k)\|^2}{\varepsilon_1}\right)\right)$ [a] | $\mathcal{O}\left((1+\gamma L_i)\log\left(\frac{1}{\varepsilon_2}\right)\right)$ |
| Accelerate gradient descent | $\mathcal{O}\left(\sqrt{1+\gamma L_i}\log\left(\frac{\|x_k-\text{prox}_{\gamma f_i}(x_k)\|^2}{\varepsilon_1}\right)\right)$ | $\mathcal{O}\left(\sqrt{1+\gamma L_i}\log\left(\frac{1}{\varepsilon_2}\right)\right)$ |

[a] We can easily provide an upper bound of $\|x_k - \text{prox}_{\gamma f_i}(x_k)\|^2$ for determining the number of local computations needed.

- Finally, we validate our theoretical findings through numerical experiments. Our numerical results suggest that the proposed technique of relative approximation effectively eliminates bias. In some cases, the algorithm even outperforms FedProx with exact updates, further validating the effectiveness of server extrapolation, even when proximal updates are inexact.

## 1.2 RELATED WORK

Arguably, stochastic gradient descent (SGD) (Robbins & Monro, 1951; Ghadimi & Lan, 2013; Gower et al., 2019; Gorbunov et al., 2020) remains one of the foundational algorithm in the field of machine learning. One can simply formulate it as

$$x_{k+1} = x_k - \eta \cdot g(x_k),$$

where $\eta > 0$ is a scalar step size, $g(x_k)$ is a possibly stochastic estimator of the true gradient $\nabla f(x_k)$. In the case when $g(x_k) = \nabla f(x_k)$, SGD becomes GD. Various extensions of SGD have been proposed since its introduction, examples include compressed gradient descent (CGD) (Alistarh et al., 2017; Khirirat et al., 2018), SGD with momentum (Loizou & Richtárik, 2017; Liu et al., 2020), SGD with matrix step size (Li et al., 2024b) and variance reduction (Gower et al., 2020; Johnson & Zhang, 2013; Gorbunov et al., 2021; Tyurin & Richtárik, 2024; Li et al., 2023). Gower et al. (2019) presented a framework for analyzing SGD with unbiased gradient estimator in the convex case based on expected smoothness. However, in practice, sometimes the gradient estimator could be biased, examples include SGD with sparsified or delayed update (Alistarh et al., 2018; Recht et al., 2011). Beznosikov et al. (2023) examined biased updates in the context of compressed gradient descent. Demidovich et al. (2024) provides a framework for analyzing SGD with biased gradient estimators in the non-convex setting.

Proximal point method (PPM) was originally introduced as a method to solve variational inequalities (Martinet, 1972; Rockafellar, 1976). The transition to the stochastic case, driven by the need to efficiently address large-scale optimization problems, leads to the development of SPPM. Due to its stability and advantage over the gradient based methods, it has been extensively studied, as documented by (Patrascu & Necoara, 2018; Bianchi, 2016; Bertsekas, 2011). For proximal algorithms to be practical, it is commonly assumed that the proximal operator can be solved efficiently, such as in cases where a closed-form solution is available. However, in large-scale machine learning models, it is rarely possible to find such a solution in closed form. To address this issue, most proximal algorithms assume that only an approximate solution is obtained, achieving a certain level of accuracy (Khaled & Jin, 2022; Sadiev et al., 2022a; Karagulyan et al., 2024). Various notions of inexactness are employed, depending on the assumptions made, the properties of the objective, and the availability of algorithms capable of efficiently finding such approximations.

Moreau envelope was first introduced to handle non-smooth functions by Moreau (1965). It is also known as the Moreau-Yosida regularization. The use of the Moreau envelope as an analytical tool to analyze proximal algorithms is not novel. Ryu & Boyd (2014) noted that running a proximal algorithm on the objective is equivalent to applying gradient methods to its Moreau envelope. Davis & Drusvyatskiy (2019) analyzed stochastic proximal point method (SPPM) for weakly convex and Lipschitz functions based on this finding. Recently, Li et al. (2024a) provided an analysis of FedProx with server-side step size in the convex case, based on the reformulation of the problem using the Moreau envelope. The role of the Moreau envelope extends beyond analyzing proximal algorithms; it has also been applied in the contexts of personalized federated learning (T Dinh et al., 2020) and meta-learning (Mishchenko et al., 2023). The mathematical properties of the Moreau envelope are relatively well understood, as documented by Jourani et al. (2014); Planiden & Wang (2019; 2016).

Projection methods initially emerged as an effective tool for solving systems of linear equations or inequalities (Kaczmarz, 1937) and were later generalized to solve the convex feasibility problem (Combettes, 1997). The parallel version of this approach involves averaging the projections of the current iterates onto all existing convex sets $\mathcal{X}_i$ to obtain the next iterate, a process that is empirically known to be accelerated by extrapolation. Numerous heuristic rules have been proposed to adaptively set the extrapolation parameter, such as those by Bauschke et al. (2006) and Pierra (1984). Only recently, the mechanism behind constant extrapolation was uncovered by Necoara et al. (2019), who developed the corresponding theoretical framework. Additionally, Li et al. (2024a) provides explanations for the effectiveness of adaptive rules, revealing the connection between the extrapolation parameter and the step size of SGD when using the Moreau envelope as the global objective.

## 2 MATHEMATICAL BACKGROUND

In this work, we are interested in the distributed optimization problem which is formulated in the following finite-sum form

$$\min_{x \in \mathbb{R}^d} \left\{ f(x) := \frac{1}{n} \sum_{i=1}^n f_i(x) \right\}, \tag{1}$$

where $x \in \mathbb{R}^d$ is the model, $n$ is the number of devices/clients, $f : \mathbb{R}^d \mapsto \mathbb{R}$ is global objective, each $f_i : \mathbb{R}^d \mapsto \mathbb{R}$ is the empirical risk of model $x$ associated with the $i$-th client. Each $f_i(x)$ often has the form

$$f_i(x) := \mathbb{E}_{\xi \sim \mathcal{D}_i} [l(x, \xi)], \tag{2}$$

where the loss function $l(x, \xi)$ represents the loss of model $x$ on data point $\xi$ over the training data $\mathcal{D}_i$ owned by client $i \in [n] := \{1, 2, \ldots, n\}$. We first give the definitions for the proximal operator and Moreau envelope, which we will be using in our analysis.

**Definition 1** (Proximal operator). *The proximal operator of an extended real-valued function $\phi : \mathbb{R}^d \mapsto \mathbb{R} \cup \{+\infty\}$ with step size $\gamma > 0$ and center $x \in \mathbb{R}^d$ is defined as*

$$\text{prox}_{\gamma\phi}(x) := \arg\min_{z \in \mathbb{R}^d} \left\{ \phi\{z\} + \frac{1}{2\gamma} \|z - x\|^2 \right\}.$$

It is well-known that for any proper, closed, and convex function $\phi$, the proximal operator with any $\gamma > 0$ returns a singleton.

**Definition 2** (Moreau envelope). *The Moreau envelope of an extended real-valued function $\phi : \mathbb{R}^d \mapsto \mathbb{R} \cup \{+\infty\}$ with step size $\gamma > 0$ and center $x \in \mathbb{R}^d$ is defined as*

$$M_\phi^\gamma(x) := \min_{z \in \mathbb{R}^d} \left\{ \phi(z) + \frac{1}{2\gamma} \|z - x\|^2 \right\}.$$

By the definition of Moreau envelope, it is easy to see that

$$M_\phi^\gamma(x) = \phi\left(\text{prox}_{\gamma\phi}(x)\right) + \frac{1}{2\gamma} \left\| x - \text{prox}_{\gamma\phi}(x) \right\|^2. \tag{3}$$

Not only are their function values related, but for any proper, closed, and convex function $\phi$, the Moreau envelope is differentiable, specifically, we have:

$$\nabla M_\phi^\gamma(x) = \frac{1}{\gamma} \left( x - \text{prox}_{\gamma\phi}(x) \right). \tag{4}$$

The above identity indicates that $\phi$ and $M_\phi^\gamma$ are intrinsically related. This relationship plays a key role in our analysis. We also need the following assumptions on $f$ and $f_i$ to carry out our analysis.

**Assumption 1** (Differentiability). *The function $f_i : \mathbb{R}^d \mapsto \mathbb{R}$ in (1) is differentiable and bounded from below for all $i \in [n]$.*

**Assumption 2** (Interpolation regime). *There exists $x_\star \in \mathbb{R}^d$ such that $\nabla f_i(x_\star) = 0$ for all $i \in [n]$.*

The same as Li et al. (2024a), we assume that we are in the interpolation regime. This situation arises in modern deep learning scenarios where the number of parameters, $d$, significantly exceeds the number of data points. For justifications, we refer the readers to Arora et al. (2019); Montanari & Zhong (2022). The motivation for this assumption stems from the parallel projection methods (5) used to solve convex feasibility problems, where the intersection of all convex sets $\mathcal{X}_i$ is assumed to be non-empty, which is precisely the interpolation assumption of each $f_i$ being the indicator function of $\mathcal{X}_i$.

$$x_{k+1} = \frac{1}{n} \sum_{i=1}^n \Pi_{\mathcal{X}_i}(x_k). \tag{5}$$

It is known that for (5), the use of extrapolation would enhance its performance both in theory and practice (Necoara et al., 2019). Since $\text{prox}_{\gamma f_i}(x_k)$ can be viewed as projection to some level set of $f_i$, it is analogous to $\Pi_{\mathcal{X}_i}(x_k)$. Therefore, it is reasonable to assume that extrapolation would be effective under the same assumption.

---

**Algorithm 1** Inexact FedExProx

1: **Parameters:** extrapolation parameter $\alpha_k = \alpha > 0$, step size for the proximal operator $\gamma > 0$, starting point $x_0 \in \mathbb{R}^d$, number of clients $n$, total number of iterations $K$, proximal solution accuracy $\varepsilon \geq 0$.
2: **for** $k = 0, 1, 2 \ldots K - 1$ **do**
3:     The server broadcasts the current iterate $x_k$ to each client
4:     Each client computes an $\varepsilon$ approximation of the solution $\tilde{x}_{i,k+1} \simeq \mathrm{prox}_{\gamma f_i}(x_k)$, and sends it back to the server
5:     The server computes

$$x_{k+1} = x_k + \alpha_k \left( \frac{1}{n} \sum_{i=1}^{n} \tilde{x}_{i,k+1} - x_k \right). \tag{8}$$

6: **end for**

---

**Assumption 3** (Individual convexity). *The function $f_i : \mathbb{R}^d \mapsto \mathbb{R}$ is convex for all $i \in [n]$. This means that for each $f_i$,*

$$0 \leq f_i(x) - f_i(y) - \langle \nabla f_i(y), x - y \rangle, \quad \forall x, y \in \mathbb{R}^d. \tag{6}$$

**Assumption 4** (Smoothness). *The function $f_i : \mathbb{R}^d \mapsto \mathbb{R}$ is $L_i$-smooth, $L_i > 0$ for all $i \in [n]$. This means that for each $f_i$,*

$$f_i(x) - f_i(y) - \langle \nabla f_i(y), x - y \rangle \leq \frac{L_i}{2} \|x - y\|^2, \quad \forall x, y \in \mathbb{R}^d. \tag{7}$$

*We will use $L_{\max}$ to denote $\max_{i \in [n]} L_i$.*

**Assumption 5** (Global strong convexity). *The function $f$ is $\mu$-strongly convex, $\mu > 0$. That is*

$$f(x) - f(y) - \langle \nabla f(y), x - y \rangle \geq \frac{\mu}{2} \|x - y\|^2, \quad \forall x, y \in \mathbb{R}^d.$$

These are all standard assumptions commonly used in convex optimization. We first present our algorithm as Algorithm 1. In the following sections, we provide the analysis of this algorithm under different definitions of inexactness, respectively in Section 3 and Section 4. Details on how these inexactness levels can be achieved by each client are provided in Section 5. Finally, numerical experiments validating our results are presented in Section 6.

## 3    ABSOLUTE APPROXIMATION IN DISTANCE

As previously suggested, we assume that each proximal operator is solved inexactly, and we need to quantify this inexactness in some way. Notice that client $i$ is required to solve the following minimization problem.

$$\min_{z \in \mathbb{R}^d} A_{k,i}^{\gamma}(z) := f_i(z) + \frac{1}{2\gamma} \|z - x_k\|^2, \tag{9}$$

where $x_k$ is the current iterate and $\gamma > 0$ is a constant. Since we have assumed each function $f_i$ is convex, $A_{k,i}^{\gamma}(z)$ is $\frac{1}{\gamma}$-strongly convex with $\mathrm{prox}_{\gamma f_i}(x_k)$ being its unique minimizer. One of the most straightforward ways to measure inexactness in this case is through the squared distance to the minimizer, leading to the following definition.

**Definition 3** (Absolute approximation). *Given a proper, closed and convex function $\phi : \mathbb{R}^d \mapsto \mathbb{R}$, and a step size $\gamma > 0$, we say that a point $y \in \mathbb{R}^d$ is an $\varepsilon_1$-approximation of $\mathrm{prox}_{\gamma\phi}(x)$, if for some $\varepsilon_1 \geq 0$,*

$$\left\| y - \mathrm{prox}_{\gamma\phi}(x) \right\|^2 \leq \varepsilon_1. \tag{10}$$

In order to analyze Algorithm 1, we first transform the update rule given in (8) in the following way,

$$
\begin{aligned}
x_{k+1} &= x_k + \alpha_k \left( \frac{1}{n} \sum_{i=1}^{n} \left( \tilde{x}_{i,k+1} - \mathrm{prox}_{\gamma f_i}(x_k) \right) + \frac{1}{n} \sum_{i=1}^{n} \mathrm{prox}_{\gamma f_i}(x_k) - x_k \right) \\
&\overset{(4)}{=} x_k - \alpha_k \cdot g(x_k),
\end{aligned}
\tag{11}
$$

where

$$g(x_k) := \underbrace{\frac{1}{n} \sum_{i=1}^{n} \gamma \nabla M_{f_i}^{\gamma}(x_k)}_{\text{Gradient}} - \underbrace{\frac{1}{n} \sum_{i=1}^{n} \left( \tilde{x}_{i,k+1} - \text{prox}_{\gamma f_i}(x_k) \right)}_{\text{Bias}}. \tag{12}$$

The above reformulation suggests that Algorithm 1 is in fact, SGD with respect to global objective $\gamma M^{\gamma}(x) := \frac{1}{n} \sum_{i=1}^{n} \gamma M_{f_i}^{\gamma}(x)$ with a biased gradient estimator. Compared to SGD with an unbiased gradient estimator, its biased counterpart is less well understood. However, we are still able to obtain the following convergence guarantee using theories for biased SGD from Demidovich et al. (2024).

**Theorem 1.** *Assume Assumption 1 (Differentiability), 2 (Interpolation Regime), 3 (Individual convexity), 4 (Smoothness) and 5 (Global strong convexity) hold. If each client computes a $\varepsilon_1$-absolute approximation $\tilde{x}_{i,k+1}$ of $\text{prox}_{\gamma f_i}(x_k)$ at every iteration, such that $\left\| \tilde{x}_{i,k+1} - \text{prox}_{\gamma f_i}(x_k) \right\|^2 \leq \varepsilon_1$. We have the following convergence guarantee for Algorithm 1: For extrapolation parameter $\alpha_k = \alpha$ satisfying $0 < \alpha \leq \frac{1}{4} \cdot \frac{1}{\gamma L_{\gamma}}$, where $\gamma$ is the step size of the proximal operator, $L_{\gamma}$ is the smoothness constant of $M^{\gamma}$. The last iterate $x_K$ satisfy*

$$\mathcal{E}_K \leq \left( 1 - \frac{\alpha \gamma \mu}{8 \left( 1 + \gamma L_{\max} \right)} \right)^K \mathcal{E}_0 + \frac{4 \varepsilon_1 \left( 1 + \gamma L_{\max} \right)}{\mu} \cdot \left( 2 \alpha L_{\gamma} + \frac{1}{\gamma} \right),$$

*where $\mathcal{E}_k = \gamma M^{\gamma}(x_k) - \gamma M_{\inf}^{\gamma}$. Specifically, when choosing $\alpha = \frac{1}{4} \cdot \frac{1}{\gamma L_{\gamma}}$, we have*

$$\Delta_K \leq \left( 1 - \frac{\mu}{32 L_{\gamma} \left( 1 + \gamma L_{\max} \right)} \right)^K \frac{L_{\gamma} \left( 1 + \gamma L_{\max} \right)}{\mu} \cdot \Delta_0 + 12 \varepsilon_1 \cdot \left( \frac{1/\gamma + L_{\max}}{\mu} \right)^2,$$

*where $\Delta_K = \| x_K - x_\star \|^2$, $x_\star$ is a minimizer of $f$.*

For the sake of brevity in the following discussion, we will use the notation $\mathcal{E}_k = \gamma M^{\gamma}(x_k) - \gamma M_{\inf}^{\gamma}$, where $M_{\inf}^{\gamma}$ denotes the infimum of $M^{\gamma}$, $\Delta_k = \| x_k - x_\star \|^2$, where $x_\star$ is a minimizer of $M^{\gamma}$. Notice that since we are in the interpolation regime, according to Fact 7, the minimizer of $M^{\gamma}$ is also a minimizer of $f$. Note that instead of converging to the exact minimizer $x_\star$, the algorithm converges to a neighborhood whose size depends on both $\varepsilon_1$ and $\gamma$; the smaller $\gamma$ is, the larger the neighborhood becomes. This can be understood intuitively: A smaller $\gamma$ means less progress is made per iteration, leading to a larger accumulated error as the total number of iterations increases. The parameter $\varepsilon_1$ can be arbitrarily large, and the convergence guarantee still holds, indicating that the theory presented is quite general. However, as $\varepsilon_1$ increases, the size of the neighborhood grows proportionally, which limits the practical significance of the result. When $\varepsilon_1 = 0$, the neighborhood diminishes, and we obtain an iteration complexity of $\tilde{\mathcal{O}} \left( \frac{L_{\gamma} \left( 1 + \gamma L_{\max} \right)}{\mu} \right)^3$, which recovers the result of Li et al. (2024a) up to a constant factor. The optimal constant extrapolation parameter is now given by $\alpha_\star = \frac{1}{4} \cdot \frac{1}{\gamma L_{\gamma}}$ which is 4 times smaller than that of Li et al. (2024a).

## 4 RELATIVE APPROXIMATION IN DISTANCE

Theorem 1 offers a general theoretical framework for understanding the behavior of Algorithm 1. However, a key challenge with Algorithm 1 which utilizes inexact proximal solutions that satisfy Definition 3, is that, unless the proximal operators are solved exactly, convergence will always be limited to a neighborhood of the solution. The underlying reason is that, as the algorithm progresses, the gradient term in the gradient estimator $g(x_k)$ diminishes, whereas the bias term remains unchanged. Building on this observation, we propose employing a different type of approximation, specifically an approximation in relative distance, as defined below.

**Definition 4** (Relative approximation). *Given a convex function $\phi : \mathbb{R}^d \mapsto \mathbb{R}$ and a stepsize $\gamma > 0$, we say that a point $y \in \mathbb{R}^d$ is a $\varepsilon_2$-relative approximation of $\text{prox}_{\gamma\phi}(x)$, if for some $\varepsilon_2 \in [0, 1)$,*

$$\left\| y - \text{prox}_{\gamma\phi}(x) \right\|^2 \leq \varepsilon_2 \cdot \left\| x - \text{prox}_{\gamma\phi}(x) \right\|^2. \tag{13}$$

---

[3] We leave out the log factor in $\tilde{\mathcal{O}}(\cdot)$ notation.

The same concept of approximations have been extensively studied and widely applied in prior research, as exemplified by Solodov & Svaiter (1999). We impose the requirement that the coefficient $\varepsilon_2$ be less than 1 to ensure that the next iterate is no worse than the current one. As we can observe, if the approximation of the solution for each proximal operator satisfies Definition 4, both the gradient term and the bias term diminish as the algorithm progresses, ensuring convergence to the exact solution. Using the theory of biased SGD, we can obtain the following theorem.

**Theorem 2.** *Assume all the assumptions mentioned in Theorem 1 also hold here. If each client only computes a $\varepsilon_2$-relative approximation $\tilde{x}_{i,k+1}$ in distance with $\varepsilon_2 < \mu^2/4L_{\max}^2$, such that $\left\|\tilde{x}_{i,k+1} - \text{prox}_{\gamma f_i}(x_k)\right\|^2 \leq \varepsilon_2 \cdot \left\|x_k - \text{prox}_{\gamma f_i}(x_k)\right\|^2$. If we are running Algorithm 1 with $\alpha_k = \alpha$ satisfying*

$$0 < \alpha \leq \frac{1}{\gamma L_\gamma} \cdot \frac{\mu - 2\sqrt{\varepsilon_2}L_{\max}}{\mu + 4\sqrt{\varepsilon_2}L_{\max} + 4\varepsilon_2 L_{\max}}.$$

*Then the iterates generated by Algorithm 1 satisfies*

$$\mathcal{E}_K \leq \left(1 - \alpha \cdot \frac{\gamma\left(\mu - 2\sqrt{\varepsilon_2}L_{\max}\right)}{4\left(1 + \gamma L_{\max}\right)}\right)^K \mathcal{E}_0.$$

*Specifically, if we choose the largest $\alpha$ possible, we have*

$$\Delta_K \leq \left(1 - \frac{\mu}{4L_\gamma\left(1 + \gamma L_{\max}\right)} \cdot S\left(\varepsilon_2\right)\right)^K \cdot \frac{L\gamma\left(1 + \gamma L_{\max}\right)}{\mu}\Delta_0,$$

*where $S(\varepsilon_2) := \frac{\left(\mu - 2\sqrt{\varepsilon_2}L_{\max}\right)\left(1 - 2\sqrt{\varepsilon_2}\frac{L_{\max}}{\mu}\right)}{\mu + 4\sqrt{\varepsilon_2}L_{\max} + 4\varepsilon_2 L_{\max}}$ satisfies $0 < S(\varepsilon_2) \leq 1$ is the factor of slowing down due to inexact proximal operator evaluation.*

Observe that when $\varepsilon_2 = 0$, meaning the proximal operators are solved exactly, the optimal extrapolation is $\alpha = \frac{1}{\gamma L_\gamma}$ and the iteration complexity is $\tilde{\mathcal{O}}\left(\frac{L_\gamma(1 + \gamma L_{\max})}{\mu}\right)$. This recovers the exact result from Li et al. (2024a). In the case of an inexact solution, as $\varepsilon_2$ increases, both $\alpha$ and $S(\varepsilon_2)$ decrease, leading to a slower rate of convergence. Note that arbitrary rough approximations are not permissible in this case, as $\varepsilon_2$ must satisfy $\varepsilon_2 = c \cdot \frac{\mu^2}{4L_{\max}^2}$, where $c < 1$.

It is worthwhile noting that Definition 4 is connected to the concept of compression. Indeed, in our case we have $x_k - \text{prox}_{\gamma f_i}(x_k) = \gamma \nabla M_{f_i}^\gamma(x_k)$, while $\tilde{x}_{i,k+1} - \text{prox}_{\gamma f_i}(x_k)$ can be interpreted as the gradient after compression, that is, $\mathcal{C}(\gamma \nabla M_{f_i}^\gamma(x_k))$. This indicates that Algorithm 1 with approximation satisfying Definition 4 can be viewed as compressed gradient descent with biased compressor. We obtain the following convergence guarantee based on theory provided by Beznosikov et al. (2023).

**Theorem 3.** *Assume all assumptions of Theorem 1 hold. Let the approximation $\tilde{x}_{i,k+1}$ all satisfies Definition 4 with $\varepsilon_2 < \mu/4L_{\max}$, that is $\left\|\tilde{x}_{i,k+1} - \text{prox}_{\gamma f_i}(x_k)\right\|^2 \leq \varepsilon_2 \cdot \left\|x_k - \text{prox}_{\gamma f_i}(x_k)\right\|^2$. If we are running Algorithm 1 with $\alpha_k = \alpha \in (0, \frac{1}{\gamma L_\gamma}]$, we have the iterates produced by it satisfying*

$$\mathcal{E}_K \leq \left(1 - \left(1 - \frac{4\varepsilon_2 L_{\max}}{\mu}\right) \cdot \frac{\gamma\mu}{4\left(1 + \gamma L_{\max}\right)} \cdot \alpha\right)^K \mathcal{E}_0.$$

*specifically, if we take the largest extrapolation ($\alpha = \frac{1}{\gamma L_\gamma} > 1$) possible, we have*

$$\Delta_K \leq \left(1 - \left(1 - \frac{4\varepsilon_2 L_{\max}}{\mu}\right) \cdot \frac{\mu}{4L_\gamma\left(1 + \gamma L_{\max}\right)}\right)^K \cdot \frac{L_\gamma\left(1 + \gamma L_{\max}\right)}{\mu}\Delta_0.$$

The convergence guarantee obtained in this way is sharper, indeed, Theorem 3 suggests that as long as $\varepsilon_2 < \mu/4L$, we are able to pick $\alpha = 1/\gamma L_\gamma$[4] which is the optimal extrapolation for exact proximal computation given in Li et al. (2024a). Notably, this implies that extrapolation is an effective technique for accelerating the algorithm in this setting, regardless of inexact proximal operator evaluations. Same as Theorem 2, the convergence is slowed down by the approximation, and in the case of $\varepsilon_2 = 0$, we recover the result in Li et al. (2024a)

---

[4]It is shown in Li et al. (2024a) that $1/\gamma L_\gamma > 1$, which justifies why $\alpha$ is called the extrapolation parameter.

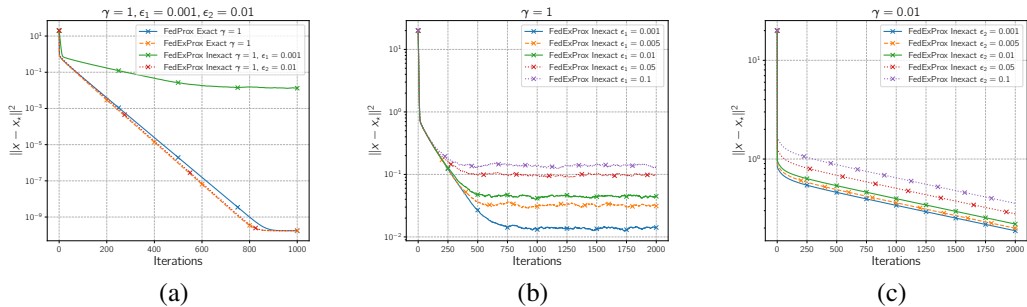

(a)                                            (b)                                            (c)

Figure 1: Comparison of FedProx, FedExProx with exact proximal evaluations, FedExProx with $\varepsilon_1$-absolute approximations for inexact proximal evaluations and FedExProx with $\varepsilon_2$-relative approximations for inexact proximal evaluations. Figure (a) presents a comparison of the four algorithms discussed above. Figure (b) illustrates the impact of different values of $\varepsilon_1$ on FedExProx with absolute approximation. Figure (c) demonstrates how varying values of $\varepsilon_2$ affect FedExProx with relative approximation.

## 5    ACHIEVING THE LEVEL OF INEXACTNESS

To fully comprehend the overall complexity of Algorithm 1, it is essential to examine whether the inexactness in evaluating the proximal operators can be effectively achieved. Since each $\text{prox}_{\gamma f_i}(x_k)$ is computed locally by the corresponding client, the client has access to all the necessary data points for the computation. Thus, the most straightforward approach is to have each client perform GD. Based on existing theories for GD, we obtain the following theorem on the local complexities.

**Theorem 4** (Local computation via GD). *Assume Assumption 1 (Differentiability), Assumption 3 (Individual convexity) and Assumption 4 (Smoothness) hold. The iteration complexity for the $i$-th client to provide an approximation using GD in the $k$-th iteration with local step size $\eta_i = \frac{\gamma}{1+\gamma L_i}$, satisfying Definition 3 is $\mathcal{O}\left((1+\gamma L_i)\log\left(\|x_k - \text{prox}_{\gamma f_i}(x_k)\|^2 / \varepsilon_1\right)\right)$, and for Definition 4, it is $\mathcal{O}\left((1+\gamma L_i)\log(1/\varepsilon_2)\right)$.*

Note that there are no constraints on $\varepsilon_1$, and since $\left\|x_k - \text{prox}_{\gamma f_i}(x_k)\right\|^2 \leq \|\gamma\nabla f(x_k)\|^2$ by (44), it is straightforward to adjust GD to optimize the approximation. However, for $\varepsilon_2$, we require $\varepsilon_2 < \frac{\mu}{4L_{\max}}$. In practice, $\varepsilon_2$ can be set to a sufficiently small value to satisfy this condition, though this will increase the number of local iterations performed by each client. The complexity bounds also indicate that as the local step size $\gamma$ increases, it becomes more challenging to compute the approximation. Alternatively, other algorithms can be employed to find such an approximation. For instance, by leveraging the structure in (2), SGD can be used as a local solver for the proximal operator when computational resources are limited. We can use the accelerated gradient descent (AGD) of Nesterov (2004) to obtain a better iteration complexity for each client.

**Theorem 5** (Local computation via AGD). *Assume all assumptions mentioned in Theorem 4 hold. The iteration complexities for the $i$-th client to provide an approximation in the $k$-the iteration using AGD with local step size $\eta_i = \frac{\gamma}{1+\gamma L_i}$ and momentum parameter $\alpha_i = \frac{\sqrt{1+\gamma L_i}-1}{\sqrt{1+\gamma L_i}+1}$, satisfying Definition 3, Definition 4 are*

$$\mathcal{O}\left(\sqrt{1+\gamma L_i}\log\left(\frac{(1+\gamma L_i)\cdot\left\|x_k - \text{prox}_{\gamma f_i}(x_k)\right\|^2}{\varepsilon_1}\right)\right); \quad \mathcal{O}\left(\sqrt{1+\gamma L_i}\log\left(\frac{1+\gamma L_i}{\varepsilon_2}\right)\right),$$

*respectively.*

## 6    EXPERIMENTS

Finally, we provide numerical evidence to support our theoretical findings. We refer the readers to Appendix H for the details of the settings and the corresponding experiments.

See Figure 1 for an overview of several experiments we conducted. In Figure 1 (a), we compare the performance of FedProx, FedExProx with exact proximal evaluations, FedExProx with $\varepsilon_1$-absolute approximations for inexact proximal evaluations, and FedExProx with $\varepsilon_2$-relative approximations for inexact proximal evaluations. Interestingly, FedExProx with relative approximations delivers strong performance when $\varepsilon_2$ is appropriately selected, and in some cases, it even outperforms FedProx with exact updates. This demonstrates the effectiveness of server extrapolation despite inexact proximal evaluations. As predicted by Theorem 1, FedExProx converges only to a neighborhood of the solution. As we Will see in Appendix H, the size of this neighborhood increases as the local step size $\gamma$ decreases, due to the accumulation of error.

In Figure 1 (b), we present a comparison of FedExProx with absolute approximations under different levels of inexactness $\varepsilon_1$. In all cases, the algorithm converges to a neighborhood of the solution, with larger inexactness resulting in a larger neighborhood.

In Figure 1 (c), we compare FedExProx with relative approximations under varying levels of inexactness $\varepsilon_2$. In all cases, the algorithm converges to the exact solution, validating the effectiveness of relative approximation in eliminating bias. As predicted by Theorem 3, larger values of $\varepsilon_2$ slow the algorithm's convergence.

## 7 CONCLUSIONS

### 7.1 LIMITATIONS

Despite achieving satisfactory results in the full-batch setting, the client sampling setting did not yield similar outcomes. This may be attributed to the nature of biased compression, which likely requires adjustments to the algorithm itself for resolution. Nonetheless, we provide the analysis in Appendix F for reference. Unlike Li et al. (2024a), the presence of bias makes it unclear how to incorporate adaptive step-size rules such as gradient diversity in our case. The only permissible inexactness for gradient diversity arises from client sub-sampling in the interpolation regime.

### 7.2 FUTURE WORK

There are still open problems to be addressed. For example, can Algorithm 1 be modified to incorporate the benefits of error feedback? Is it possible to eliminate the interpolation regime assumption while still demonstrating that extrapolation is theoretically beneficial for FedExProx? Another direction that may be of independent interest is to develop adaptive rules of determining the step size for SGD with biased update.

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

CONTENTS

## A  NOTATIONS

Throughout the paper, we use the notation $\|\cdot\|$ to denote the standard Euclidean norm defined on $\mathbb{R}^d$ and $\langle \cdot, \cdot \rangle$ to denote the standard Euclidean inner product. Given a differentiable function $f : \mathbb{R}^d \mapsto \mathbb{R}$, its gradient is denoted as $\nabla f(x)$. We use the notation $D_f(x, y)$ to denote the Bregman divergence associated with a function $f : \mathbb{R}^d \mapsto \mathbb{R}$ between $x$ and $y$. The notation $\inf f$ is used to denote the minimum of a function $f : \mathbb{R}^d \mapsto \mathbb{R}$. We use $\text{prox}_{\gamma\phi}(x)$ to denote the proximity operator of function $\phi : \mathbb{R}^d \mapsto \mathbb{R}$ with $\gamma > 0$ at $x \in \mathbb{R}^d$, and $M_\phi^\gamma(x)$ to denote the corresponding Moreau Envelope. We denote the average of the Moreau envelope of each local objective $f_i$ by the notation $M^\gamma : \mathbb{R}^d \mapsto \mathbb{R}$. Specifically, we define $M^\gamma(x) = \frac{1}{n} \sum_{i=1}^n M_f^\gamma(x)$. Note that $M^\gamma(x)$ has an implicit dependence on $\gamma$, its smoothness constant is denoted by $L_\gamma$. We say an extended real-valued function $f : \mathbb{R}^d \mapsto \mathbb{R} \cup \{+\infty\}$ is proper if there exists $x \in \mathbb{R}^d$ such that $f(x) < +\infty$. We say an extended real-valued function $f : \mathbb{R}^d \mapsto \mathbb{R} \cup \{+\infty\}$ is closed if its epigraph is a closed set. We use the notation $\mathcal{E}_k = \gamma M^\gamma(x_k) - \gamma M_{\inf}^\gamma$ to denote the function value suboptimality of $\gamma M^\gamma$ at $x_k$, and $\Delta_k = \|x_k - x_\star\|^2$ to denote the squared distance. The notation $\mathcal{O}(\cdot)$ is used to describe complexity while omitting constant factors, whereas $\tilde{\mathcal{O}}(\cdot)$ is used when logarithmic factors are also omitted. For approximation $y \in \mathbb{R}^d$ of $\text{prox}_{\gamma f}(x)$, we use $\varepsilon_1$ as the accuracy of absolute approximation such that $\|y - \text{prox}_{\gamma f}(x)\|^2 \leq \varepsilon_1$, and we use $\varepsilon_2$ as the accuracy of relative approximation such that $\|y - \text{prox}_{\gamma f}(x)\|^2 \leq \varepsilon_2 \cdot \|x - \text{prox}_{\gamma f}(x)\|^2$.

## B  FACTS AND LEMMAS

**Fact 1** (Young's inequality). *For any two vectors $x, y \in \mathbb{R}^d$, the following inequality holds,*

$$\|x + y\|^2 \leq 2\|x\|^2 + 2\|y\|^2. \tag{14}$$

**Fact 2** (Property of convex smooth functions). *Let $\phi : \mathbb{R}^d \mapsto \mathbb{R}$ be differentiable. The following statements are equivalent:*

1. *$\phi$ is convex and $L$-smooth.*

2. *$0 \leq 2D_\phi(x, y) \leq L\|x - y\|^2$ for all $x, y \in \mathbb{R}^d$.*

3. *$\frac{1}{L}\|\nabla\phi(x) - \nabla\phi(y)\|^2 \leq 2D_\phi(x, y)$ for all $x, y \in \mathbb{R}^d$.*

*The notation $D_\phi(x, y)$ denotes the Bregman divergence associate with $\phi$ at $x, y \in R^d$, defined as*

$$D_\phi(x, y) = \phi(x) - \phi(y) - \langle \nabla\phi(y), x - y \rangle.$$

The following two facts establish that the convexity and smoothness of a function $\phi : \mathbb{R}^d \mapsto \mathbb{R}$ ensure the convexity and smoothness of its Moreau envelope.

**Fact 3** (Convexity of Moreau envelope). *(Beck, 2017, Theorem 6.55) Let $\phi : \mathbb{R}^d \mapsto \mathbb{R} \cup \{+\infty\}$ be a proper and convex function. Then $M_\phi^\gamma$ is a convex function.*

**Fact 4** (Smoothness of Moreau envelope). *(Li et al., 2024a, Lemma 4) Let $\phi : \mathbb{R}^d \mapsto \mathbb{R}$ be a convex and $L$-smooth function. Then $M_\phi^\gamma$ is $\frac{L}{1+\gamma L}$-smooth.*

The following fact illustrates the relationship between the minimizer of a function $\phi$ and its Moreau envelope $M_\phi^\gamma$.

**Fact 5** (Minimizer equivalence). *(Li et al., 2024a, Lemma 5) Let $\phi : \mathbb{R}^d \mapsto \mathbb{R} \cup \{+\infty\}$ be a proper, closed and convex function. Then for any $\gamma > 0$, $\phi$ and $M_\phi^\gamma$ has the same set of minimizers.*

In our case, we assume each $f_i$ from (1) is convex and $L_i$-smooth. Therefore by Fact 3 and Fact 4, we know that each $M_{f_i}^\gamma$ is also convex and $\frac{L_i}{1+\gamma L_i}$-smooth. This means that $M_\gamma = \frac{1}{n} \sum_{i=1}^n M_{f_i}^\gamma$ is also convex and smooth. We denote its smoothness constant as $L_\gamma$, and the following fact provides a range for this constant.

**Fact 6** (Global convexity and smoothness). *(Li et al., 2024a, Lemma 7) Let each $f_i$ be proper, closed convex and $L_i$-smooth. Then $M^\gamma$ is convex and $L_\gamma$-smooth with*

$$\frac{1}{n^2} \sum_{i=1}^n \frac{L_i}{1+\gamma L_i} \leq L_\gamma \leq \frac{1}{n} \sum_{i=1}^n \frac{L_i}{1+\gamma L_i}.$$

The following fact establishes that the minimizer of $f$ and $M^\gamma$ are the same.

**Fact 7** (Global minimizer equivalence). *(Li et al., 2024a, Lemma 8) If we let every $f_i : \mathbb{R}^d \mapsto \mathbb{R} \cup \{+\infty\}$ be proper, closed and convex, then $f(x) = \frac{1}{n} \sum_{i=1}^n f_i(x)$ has the same set of minimizers and minimum as*

$$M^\gamma(x) = \frac{1}{n} \sum_{i=1}^n M_{f_i}^\gamma(x),$$

*if we are in the interpolation regime and $0 < \gamma < \infty$.*

The above fact demonstrates that running SGD on the objective $M^\gamma$ will lead us to the correct destination, as the minimizers of $M^\gamma$ and $f$ are identical in our setting. In problem (1), if we assume that $f$ is strongly convex, then we have $M^\gamma$ satisfies the following star strong convexity inequality.

**Fact 8** (Star strong convexity). *(Li et al., 2024a, Lemma 11) Assume Assumption 1 (Differentiability), Assumption 2 (Interpolation Regime), Assumption 3 (Individual convexity), Assumption 4 (Smoothness) and Assumption 5 (Global strong convexity) hold, then the convex function $M^\gamma(x)$ satisfies the following inequality,*

$$M^\gamma(x) - M_{\inf}^\gamma \geq \frac{\mu}{1+\gamma L_{\max}} \cdot \frac{1}{2} \|x - x_\star\|^2,$$

*for any $x \in \mathbb{R}^d$ and a minimizer $x_\star$ of $M^\gamma(x)$.*

The above fact implies that the strong convexity of $f$ translates to the star strong convexity of $M^\gamma$. Star strong convexity is also known as quadratic growth (QG) condition (Anitescu, 2000). In the case of a convex function, it is also known as optimal strong convexity (Liu & Wright, 2015) and semi-strong convexity (Gong & Ye, 2014). It is known that for a convex function satisfying quadratic growth condition, it also satisfies the Polyak-Lojasiewicz inequality (Polyak, 1964) which is described by the following lemma. Notice that since Algorithm 1 can be viewed as running SGD with objective $\gamma M^\gamma$ and a fixed step size $\alpha_k = \alpha$, we describe the inequality based on $\gamma M^\gamma$ in the following lemma.

**Lemma 1** (PL-inequality). *Let Assumption 1 (Differentiability), Assumption 2 (Interpolation Regime), Assumption 3 (Individual convexity), Assumption 4 (Smoothness) and Assumption 5 (Global strong convexity) hold, then $\gamma M^\gamma(x)$ satisfies the following Polyak-Lojasiewicz inequality,*

$$\|\gamma \nabla M^\gamma(x)\|^2 \geq 2 \cdot \frac{\gamma \mu}{4(1+\gamma L_{\max})} (\gamma M^\gamma(x) - \gamma M_{\inf}^\gamma), \tag{15}$$

*where $x \in \mathbb{R}^d$ is an arbitrary vector and $x_\star$ is a minimizer of $M^\gamma(x)$.*

## C    THEORY OF BIASED SGD

For completeness, we provide the theory of biased SGD we used to analyze our algorithm in this paper. It is adapted from Demidovich et al. (2024), which offers a comprehensive study of various assumptions employed in the analysis of SGD with biased gradient updates. In addition, the authors introduced a new set of assumptions, referred to as the Biased ABC assumption, which are less

restrictive than all previous assumptions. The authors provided convergence guarantees for SGD with biased gradient updates in the non-convex and convex setting. Specifically, they considered the case of minimizing a function $f : \mathbb{R}^d \mapsto \mathbb{R}$,

$$\min_{x \in \mathbb{R}^d} f(x),$$

with

$$x_{k+1} = x_k - \eta g(x_k), \qquad \text{(biased SGD)}$$

where $\eta > 0$ is the stepsize, $g(x_k)$ is a possibly stochastic and biased gradient estimator. They introduced the biased ABC assumption,

**Assumption 6** (Biased-ABC). *(Demidovich et al., 2024, Assumption 9) There exists constants $A, B, C, b, c \geq 0$ such that the gradient estimator $g(x)$ for every $x \in \mathbb{R}^d$ satisfies*

$$\langle \nabla f(x), \mathbb{E}\left[g(x)\right] \rangle \geq b \left\| \nabla f(x) \right\|^2 - c$$
$$\mathbb{E}\left[ \left\| g(x) \right\|^2 \right] \leq 2A \left( f(x) - f_{\inf} \right) + B \left\| \nabla f(x) \right\|^2 + C.$$

A convergence guarantee was provided for biased SGD under Assumption 6 given that $f$ is $\widehat{L}$-smooth and $\widehat{\mu}$-PL, that is, there exists $\widehat{\mu} > 0$, such that

$$\left\| \nabla f(x) \right\|^2 \geq 2\widehat{\mu} \left( f(x) - f_{\inf} \right),$$

for all $x \in \mathbb{R}^d$.

**Theorem 6** (Theory of biased SGD). *(Demidovich et al., 2024, Theorem 4) Let $f$ be $\widehat{L}$-smooth and $\widehat{\mu}$-PL and Assumption 6 hold. If we choose a step size $\eta$ satisfying*

$$0 < \eta < \min \left\{ \frac{\widehat{\mu}b}{\widehat{L}\left(A + \widehat{\mu}B\right)}, \frac{1}{\widehat{\mu}b} \right\}. \qquad (16)$$

*Then we have*

$$\mathbb{E}\left[f(x_k) - f_{\inf}\right] \leq \left(1 - \eta\widehat{\mu}b\right)^k \left(f(x_0) - f_{\inf}\right) + \frac{LC\eta}{2\widehat{\mu}b} + \frac{c}{\widehat{\mu}b}.$$

*Under the special case of*

$$\frac{\widehat{\mu}b}{\widehat{L}\left(A + \widehat{\mu}B\right)} < \frac{1}{\widehat{\mu}b},$$

*The range of the step size can be simplified to*

$$0 < \eta \leq \frac{\widehat{\mu}b}{\widehat{L}\left(A + \widehat{\mu}B\right)},$$

*and if we take the largest possible step size, we have*

$$\mathbb{E}\left[f(x_k) - f_{\inf}\right] \leq \left(1 - \frac{\widehat{\mu}^2 b^2}{\widehat{L}\left(A + \widehat{\mu}B\right)}\right)^k \left(f(x_0) - f_{\inf}\right) + \frac{LC}{2\widehat{L}\left(A + \widehat{\mu}B\right)} + \frac{c}{\widehat{\mu}b}.$$

The constants $C, c$ determine whether the algorithm is converging to the exact solution or just a neighborhood. For $g(x) = \nabla f(x)$, clearly we have $A = 0, B = 1, b = 1, C = 0, c = 0$, and there is no neighborhood. This is expected because the algorithm reduces to standard GD The iteration complexity is give by $\tilde{\mathcal{O}}\left(\frac{\widehat{L}}{\widehat{\mu}}\right)$, which is also expected for GD.

## D  THEORY OF BIASED COMPRESSION

In this section, we present the theory of SGD with biased compression. The theory is adapted from Beznosikov et al. (2023). The authors introduced theory for analyzing compressed gradient descent (CGD) with biased compressor, both in the single node case and in the distributed case when the objective function is assumed to be strongly convex. Here, we are only concerned with the single

node case because distributed compressed gradient descent (DCGD) with biased compressor may fail to converge. To address this issue, error feedback mechanism (Seide et al., 2014; Karimireddy et al., 2019; Richtárik et al., 2021) is needed. In the single node case, the authors considered solving

$$\min_{x \in \mathbb{R}^d} f(x),$$

where $f : \mathbb{R}^d \mapsto \mathbb{R}$ is $\widehat{L}$-smooth and $\widehat{\mu}$-strongly convex, with the following compressed gradient descent algorithm

$$x_{k+1} = x_k - \eta \mathcal{C}\left(\nabla f(x_k)\right), \tag{CGD}$$

where $\mathcal{C} : \mathbb{R}^d \mapsto \mathbb{R}$ are potentially biased compression operators, $\eta > 0$ is a step size. The author proved that if certain conditions on $\mathcal{C}$ is satisfied, a corresponding convergence guarantee can then be established. Three classes of compressor/mapping were introduced.

**Definition 5** (Class $\mathbb{B}^1$)**.** *We say a mapping $\mathcal{C} \in \mathbb{B}^1\left(\alpha, \beta\right)$ for some $\alpha, \beta > 0$ if*

$$\alpha \left\|x\right\|^2 \leq \mathbb{E}\left[\left\|\mathcal{C}\left(x\right)\right\|^2\right] \leq \beta \left\langle \mathbb{E}\left[\mathcal{C}\left(x\right)\right], x \right\rangle, \qquad \forall x \in \mathbb{R}^d.$$

**Definition 6** (Class $\mathbb{B}^2$)**.** *We say a mapping $\mathcal{C} \in \mathbb{B}^2\left(\xi, \beta\right)$ for some $\xi, \beta > 0$ if*

$$\max\left\{\xi \left\|x\right\|^2, \frac{1}{\beta}\mathbb{E}\left[\left\|\mathcal{C}\left(x\right)\right\|^2\right]\right\} \leq \left\langle \mathbb{E}\left[\mathcal{C}\left(x\right)\right], x \right\rangle, \qquad \forall x \in \mathbb{R}^d.$$

**Definition 7** (Class $\mathbb{B}^3$)**.** *We say a mapping $\mathcal{C} \in \mathbb{B}^3\left(\delta\right)$ for some $\delta > 0$, if*

$$\mathbb{E}\left[\left\|\mathcal{C}\left(x\right) - x\right\|^2\right] \leq \left(1 - \frac{1}{\delta}\right) \left\|x\right\|^2.$$

The authors proved the following theorem about the convergence of the algorithm, the notation $\mathcal{F}_k$ is used to denote $\mathbb{E}\left[f(x_k)\right] - f_{\inf}$, with $\mathcal{F}_0 = f(x_0) - f_{\inf}$,

**Theorem 7.** *Let $\mathcal{C} \in \mathbb{B}^1\left(\alpha, \beta\right)$. Then we have $\mathcal{F}_k \leq \left(1 - {}^{\alpha}\!/\!{}_{\beta}\eta\widehat{\mu}\left(2 - \eta\beta\widehat{L}\right)\right)\mathcal{F}_{k-1}$, as long as $0 \leq \eta \leq \frac{2}{\beta\widehat{L}}$. If we choose $\eta = \frac{1}{\beta\widehat{L}}$, we have*

$$\mathcal{F}_k \leq \left(1 - \frac{\alpha}{\beta^2} \cdot \frac{\widehat{\mu}}{\widehat{L}}\right)^K \mathcal{F}_0. \tag{17}$$

*Let $\mathcal{C} \in \mathbb{B}^2\left(\xi, \beta\right)$. Then we have $\mathcal{F}_k \leq \left(1 - \xi\eta\left(2 - \eta\beta\right)\widehat{L}\right)\mathcal{F}_{k-1}$, as long as $0 \leq \eta \leq \frac{2}{\beta\widehat{L}}$. If we choose $\eta = \frac{1}{\beta\widehat{L}}$, we have*

$$\mathcal{F}_k \leq \left(1 - \frac{\xi}{\beta} \cdot \frac{\widehat{\mu}}{\widehat{L}}\right)^k \mathcal{F}_0. \tag{18}$$

*Let $\mathcal{C} \in \mathbb{B}^3\left(\delta\right)$. Then we have $\mathcal{F}_k \leq \left(1 - \frac{1}{\delta}\eta\widehat{\mu}\right)\mathcal{F}_{k-1}$, as long as $0 \leq \eta \leq \frac{1}{\widehat{L}}$. If we choose $\eta = \frac{1}{\widehat{L}}$, we have*

$$\mathcal{F}_k \leq \left(1 - \frac{1}{\delta} \cdot \frac{\widehat{\mu}}{\widehat{L}}\right)^k \mathcal{F}_0. \tag{19}$$

Notice that when $\mathcal{C}\left(x\right) = x$, that is, when no compression happens, we have $\alpha = \beta = \xi = \delta = 1$. In this case, the iteration complexity of CGD is given by $\tilde{\mathcal{O}}\left(\frac{\widehat{L}}{\widehat{\mu}}\right)$ and we recover the result of GD. It is worth noting that Theorem 7 remains valid if the condition of $f$ being $\widehat{\mu}$-strongly convex is replaced with $f$ being $\widehat{\mu}$-PL.

## E    DISCUSSION OF USED ASSUMPTIONS

In this section, we provide a discussion of the assumptions used in the paper.

**Convexity:**   The motivation behind FedExProx stems from the parallel projection method Combettes (1997) of solving the convex feasibility problem. Initially, it was observed that extrapolation can accelerate the parallel projection method (in this convex interpolation setting). Given the similarity between projection operators and proximal operators (the latter can be viewed as a projection to a level set of the function), the FedExProx algorithm was developed. In this context, extrapolation is considered in conjunction with convexity; whether it remains beneficial in non-convex settings is still unclear. This rationale led us to focus on the convex case first.

**Smoothness:**   The smoothness assumption Assumption 4 is pretty common in convex optimization, and we adopt it here for simplicity of discussion and presentation. In fact, even if we do not assume each local objective function $f_i$ to be $L_i$-smooth, the corresponding Moreau envelope $M_{f_i}^\gamma$ is still $\frac{1}{\gamma}$-smooth as illustrated in Li et al. (2024a). Consequently, the inexact FedExProx still yields a form of SGD with a biased gradient estimator on the convex smooth objective $M^\gamma$. This allows us to leverage the relevant theoretical framework to analyze the convergence result in this scenario. Although some technical nuances arise, they do not impact the validity of our conclusion.

**Interpolation regime:**   Notice that, FedProx itself does not require the interpolation regime assumption. However, like FedExProx and its inexact variant, it converges to a neighborhood of the solution rather than the exact solution. The interpolation assumption was initially introduced based on the motivation behind FedExProx. It is known that the parallel projection method for solving convex feasibility problems is accelerated by extrapolation. Given the similarity between projection operators to convex sets and proximal operators of convex functions (which are, in fact, projections onto certain level sets of the function), FedExProx was proposed. The interpolation assumption here corresponds to the assumption that the intersection of these convex sets is non-empty in the convex feasibility problem. Although this assumption may seem somewhat arbitrary in the context of FedProx, it feels more intuitive when considering FedExProx through the lens of the parallel projection method. In the absence of the interpolation regime assumption, the algorithm will converge to a neighborhood of the true minimizer, $x_\star$, of $f$. This occurs because $f$ and $M^\gamma$ are guaranteed to share the same minimizer only under the interpolation regime assumption, as established in Fact 7. Since inexact FedExProx can be formulated as SGD with a biased gradient estimator on the objective $M^\gamma = \frac{1}{n} \sum_{i=1}^n M_{f_i}^\gamma$, it converges to the minimizer $x_\star'$, provided that inexactness is properly bounded. As a result, the algorithm converges to $x_\star'$, located within a $\|x_\star - x_\star'\|$-neighborhood of $x_\star$. Notably, the effects of inexactness and interpolation are, in some sense, "orthogonal", meaning they do not interfere with each other.

**Global strong convexity:**   Notice that we do not assume each function $f_i$ is strongly convex, but rather, the global objective $f$ is strongly convex. This is for the simplicity of presentation and discussion. One may consider extend the algorithm into the general convex case. To establish a convergence guarantee, one may notice that in the general convex case, FedExProx still results in biased SGD on the Moreau envelope objective $M^\gamma$ in the general convex and smooth case. The specific approximation used in the algorithm allows for the application of various existing tools for biased SGD. Biased SGD has been extensively studied in recent years; for example, Demidovich et al. (2024) provides a comprehensive overview of its analysis across different settings. Depending on the assumptions, one can adopt different theoretical frameworks to analyze FedExProx, as it is effectively equivalent to biased SGD applied to the envelope objective. For more details on those assumptions, we refer the readers to Demidovich et al. (2024). In our work, we demonstrate that the theory of biased compression provides a tighter convergence guarantee for relative approximation. However, existing theories for biased compression are limited to the strongly convex case, and extending them to the stochastic setting offers no advantages due to the bias introduced. To generalize this approach to a broader context, incorporating error feedback alongside biased compression is a promising direction. This, however, necessitates modifications to the original algorithm, which we leave as a future work.

# F  ANALYSIS OF INEXACT FEDEXPROX IN THE CLIENT SAMPLING SETTING

In this section, we will discuss the case where we do client sampling in algorithm 1, we first formulate the algorithm as below. For the sake of simplicity, we use $\tau$-nice sampling as an example.

---

**Algorithm 2** Inexact FedExProx with $\tau$-nice sampling

---

1: **Parameters:** extrapolation parameter $\alpha_k = \alpha > 0$, step size for the proximal operator $\gamma > 0$, starting point $x_0 \in \mathbb{R}^d$, number of clients $n$, size of minibatch $\tau$, total number of iterations $K$, proximal solution accuracy $\varepsilon_2 \geq 0$.
2: **for** $k = 0, 1, 2 \ldots K - 1$ **do**
3:     The server broadcasts the current iterate $x_k$ to a selected set of client $S_k$ of size $\tau$
4:     Each selected client computes a $\varepsilon$ approximation of the solution $\tilde{x}_{i,k+1} \simeq \mathrm{prox}_{\gamma f_i}(x_k)$, and sends it back to the server
5:     The server computes

$$x_{k+1} = x_k + \alpha_k \left( \frac{1}{\tau} \sum_{i \in S_k} \tilde{x}_{i,k+1} - x_k \right). \tag{20}$$

6: **end for**

---

## F.1  RELATIVE APPROXIMATION IN DISTANCE

**The failure of biased compression theory:**  Similar to Theorem 7, we initially apply the theory from Beznosikov et al. (2023), as it provides improved results in the full-batch scenario. We first define the compressing mapping $\mathcal{C}_\tau$ in this case,

$$\mathcal{C}_\tau \left( \gamma \nabla M^\gamma(x_k) \right) = \frac{1}{\tau} \sum_{i \in S_k} \left( \gamma \nabla M_{f_i}^\gamma(x_k) - \left( \tilde{x}_{i,k+1} - \mathrm{prox}_{\gamma f_i}(x_k) \right) \right). \tag{21}$$

One can verify for every $x_k$ and $\varepsilon_2$-approximation $\tilde{x}_{i,k+1}$ of $\mathrm{prox}_{\gamma f_i}(x_k)$, we have

$$\mathcal{C}_\tau \in \mathbb{B}^3 \left( \delta = \frac{\mu}{\mu - 4\varepsilon_2 L_{\max} - \frac{n-\tau}{\tau(n-1)} \left[ 4 \left( 2 + \varepsilon_2 \right) L_{\max} - 2\mu \right]} \right)$$

In the case of $\tau = n$, we have $\mathcal{C}_n \in \mathbb{B}^3 \left( \frac{\mu}{\mu - 4\varepsilon_2 L_{\max}} \right)$, which recovers the result of (42). When $\tau = 1, \varepsilon_2 = 0$, however, this is problematic, as $\mathcal{C}_1 \in \mathbb{B}^3 \left( \delta = \frac{\mu}{3\mu - 8L_{\max}} \right)$. Notice that we require $\delta > 0$, so we require $3\mu > 8L_{\max}$ which only holds in a very restrictive setting. This is due to the stochasticity contained in (21), which arises from client sampling.

**Theory of biased SGD:**  The algorithm does converge, however, and one can use the theory of Demidovich et al. (2024) to obtain a convergence guarantee.

**Theorem 8.** *Assume Assumption 1 (Differentiability), Assumption 2 (Interpolation regime), Assumption 3 (Individual convexity), Assumption 4 (Smoothness) and Assumption 5 (Global strong convexity) hold. Let the approximation $\tilde{x}_{i,k+1}$ all satisfies Definition 4 with $\varepsilon_2 < \frac{\mu^2}{4L_{\max}^2}$, that is*

$$\left\| \tilde{x}_{i,k+1} - \mathrm{prox}_{\gamma f_i}(x_k) \right\|^2 \leq \varepsilon_2 \cdot \left\| x_k - \mathrm{prox}_{\gamma f_i}(x_k) \right\|^2,$$

*holds for all client $i$ at iteration $k$. If we are running Algorithm 2 with minibatch size $\tau$ and extrapolation parameter $\alpha_k = \alpha > 0$ satisfying*

$$\alpha \leq \frac{1}{\gamma L_\gamma} \cdot \frac{\mu - 2\sqrt{\varepsilon_2} L_{\max}}{\mu + 4\varepsilon_2 L_{\max} + 4\sqrt{\varepsilon_2} L_{\max} + \frac{n-\tau}{\tau(n-1)} \cdot \left( 4L_{\max} + 4\sqrt{\varepsilon_2} L_{\max} - \mu \right)}$$

*Then the iterates generated by Algorithm 2 satisfies*

$$\mathbb{E}\left[ \mathcal{E}_K \right] \leq \left( 1 - \alpha \cdot \frac{\gamma \left( \mu - 2\sqrt{\varepsilon_2} L_{\max} \right)}{4 \left( 1 + \gamma L_{\max} \right)} \right)^K \mathcal{E}_0. \tag{22}$$

*Specifically, if we choose the largest $\alpha$ possible, we have*

$$\mathbb{E}\left[\Delta_K\right] \leq \left(1 - \frac{\mu}{4L_\gamma\left(1 + \gamma L_{\max}\right)} \cdot S\left(\varepsilon_2, \tau\right)\right)^K \cdot \frac{L\gamma\left(1 + \gamma L_{\max}\right)}{\mu} \Delta_0,$$

*where $S\left(\varepsilon_2, \tau\right)$ is defined as*

$$S\left(\varepsilon_2, \tau\right) := \frac{\left(\mu - 2\sqrt{\varepsilon_2}L_{\max}\right)\left(1 - 2\sqrt{\varepsilon_2}\frac{L_{\max}}{\mu}\right)}{\mu + 4\varepsilon_2 L_{\max} + 4\sqrt{\varepsilon_2}L_{\max} + \frac{n-\tau}{\tau(n-1)} \cdot \left(4L_{\max} + 4\sqrt{\varepsilon_2}L_{\max} - \mu\right)},$$

*satisfying*

$$0 < S\left(\varepsilon_2, \tau\right) \leq 1.$$

Notice that we have $S\left(\varepsilon_2, \tau = n\right) = S\left(\varepsilon_2\right)$, which appears in Theorem 2. For the special case when $\varepsilon_2 = 0$, every proximal operator is solved exactly. The range of $\alpha$ becomes,

$$0 < \alpha \leq \frac{1}{\gamma L_\gamma} \cdot \frac{\mu}{\frac{n-\tau}{\tau(n-1)} \cdot 4L_{\max} + \frac{n(\tau-1)}{\tau(n-1)}\mu}.$$

According to Li et al. (2024a),

$$0 < \alpha \leq \frac{1}{\gamma L_\gamma} \cdot \frac{L_\gamma\left(1 + \gamma L_{\max}\right)}{\frac{n-\tau}{\tau(n-1)}L_{\max} + \frac{n(\tau-1)}{\tau(n-1)} \cdot L_\gamma\left(1 + \gamma L_{\max}\right)}.$$

Clearly the bound we obtain here is suboptimal, since we have $\mu \leq L_\gamma\left(1 + \gamma L_{\max}\right)$ according to (27). This is due to the previously mentioned issue: the nature of biased compression. When client sampling is used together with biased compressors, it does not necessarily guarantee any benefits. To solve this, the modification of the algorithm itself may be needed, which we consider as a future work direction.

### F.2 ABSOLUTE APPROXIMATION IN DISTANCE

Similarly to Theorem 8, by applying the theory of biased SGD (Demidovich et al., 2024), we can derive a convergence guarantee for the minibatch case, though with a suboptimal convergence rate. For brevity and clarity, we do not include the details here.

## G PROOF OF THEOREMS AND LEMMAS

### G.1 PROOF OF LEMMA 1

Using Fact 8, we have

$$M^\gamma\left(x\right) - M_{\inf}^\gamma \geq \frac{\mu}{1 + \gamma L_{\max}} \cdot \frac{1}{2}\|x - x_\star\|^2, \tag{23}$$

where $x \in \mathbb{R}^d$ is any vector, $x_\star$ is a minimizer of $M^\gamma$, by Fact 5, it is also a minimizer of $f$. Since we assume each function $f_i$ is convex, by Fact 3, we know that $M_{f_i}^\gamma$ is also convex. As a result, the average of $M_{f_i}^\gamma$, $M^\gamma$ is also a convex function. Utilizing the convexity of $M^\gamma$, we have,

$$M_{\inf}^\gamma \geq M^\gamma\left(x\right) + \left\langle\nabla M^\gamma\left(x\right), x_\star - x\right\rangle.$$

Rearranging terms we get,

$$\left\langle\nabla M^\gamma\left(x\right), x - x_\star\right\rangle \geq M^\gamma\left(x\right) - M_{\inf}^\gamma. \tag{24}$$

As a result, we have

$$\left\langle\nabla M^\gamma\left(x\right), x - x_\star\right\rangle \overset{(23)+(24)}{\geq} \frac{\mu}{1 + \gamma L_{\max}} \cdot \frac{1}{2}\|x - x_\star\|^2.$$

Using Cauchy-Schwarz inequality, we have

$$\|\nabla M^\gamma(x)\| \|x - x_\star\| \geq \langle \nabla M^\gamma(x), x - x_\star \rangle \geq \frac{\mu}{1 + \gamma L_{\max}} \cdot \frac{1}{2} \|x - x_\star\|^2.$$

When $\|x - x_\star\| > 0$, the above inequality leads to

$$\|\nabla M^\gamma(x)\| \geq \frac{\mu}{2(1 + \gamma L_{\max})} \cdot \|x - x_\star\|, \tag{25}$$

which also holds when $\|x - x_\star\| = 0$. Now using (24) and (25), we obtain

$$\begin{aligned}
M^\gamma(x) - M_{\inf}^\gamma &\overset{(24)}{\leq} \langle \nabla M^\gamma(x), x - x_\star \rangle \\
&\leq \|\nabla M^\gamma(x)\| \|x - x_\star\| \\
&\overset{(25)}{\leq} \frac{2(1 + \gamma L_{\max})}{\mu} \|\nabla M^\gamma(x)\|^2.
\end{aligned}$$

A simple rearranging of terms result in

$$\|\gamma \nabla M^\gamma(x)\|^2 \geq 2 \cdot \frac{\gamma \mu}{4(1 + \gamma L_{\max})} (\gamma M^\gamma(x) - \gamma M_{\inf}^\gamma).$$

Up till here we have already proved the statement in the lemma, but we want to look at the strongly constant $\mu$ of $f$ a little bit. In order to provide an upper bound of $\mu$, we notice that due to Fact 4, each $M_{f_i}^\gamma$ is $\frac{L_i}{1 + \gamma L_i}$-smooth and therefore $M^\gamma$ is smooth. We use the notation $L_\gamma$ to denote its smoothness constant. Applying the smoothness of $M^\gamma(x)$, we have

$$M^\gamma(x) \leq M^\gamma(x_\star) + \langle \nabla M^\gamma(x_\star), x - x_\star \rangle + \frac{L_\gamma}{2} \|x - x^\star\|^2.$$

Utilizing the fact that $\nabla M^\gamma(x_\star) = 0$, we have

$$M^\gamma(x) - M_{\inf}^\gamma \leq \frac{L_\gamma}{2} \|x - x_\star\|^2 \tag{26}$$

Combining (26) and (23), we can deduce that

$$\frac{\mu}{1 + \gamma L_{\max}} \cdot \frac{1}{2} \|x - x_\star\|^2 \leq M^\gamma(x) - M_{\inf}^\gamma \leq \frac{L_\gamma}{2} \|x - x_\star\|^2.$$

which results in the estimate that

$$\mu \leq L_\gamma(1 + \gamma L_{\max}). \tag{27}$$

## G.2 PROOF OF THEOREM 1

Let us first recall that after reformulation, Algorithm 1 can be written as

$$x_{k+1} = x_k - \alpha \cdot g(x_k),$$

where $g(x_k)$ is defined as

$$g(x_k) := \frac{1}{n} \sum_{i=1}^n \gamma \nabla M_{f_i}^\gamma(x_k) - \frac{1}{n} \sum_{i=1}^n \left( \tilde{x}_{i,k+1} - \text{prox}_{\gamma f_i}(x_k) \right).$$

We view this as running full batch biased SGD with stepsize $\alpha$ and global objective $\gamma M^\gamma(x)$. We first examine if Assumption 6 (Biased-ABC) holds for arbitrary $x_k$. Since we are in the full batch case, it is easy to see that

$$\mathbb{E}[g(x_k)] = g(x_k).$$

Since our objective now is $\gamma M^\gamma(x)$, we have that

$$\begin{aligned}
\langle \gamma \nabla M^\gamma(x_k), g(x_k) \rangle &= \left\langle \gamma \nabla M^\gamma(x_k), \gamma \nabla M^\gamma(x_k) - \frac{1}{n} \sum_{i=1}^n \left( \tilde{x}_{i,k+1} - \text{prox}_{\gamma f_i}(x_k) \right) \right\rangle \\
&= \|\gamma \nabla M^\gamma(x_k)\|^2 - \underbrace{\left\langle \gamma \nabla M^\gamma(x_k), \frac{1}{n} \sum_{i=1}^n \left( \tilde{x}_{i,k+1} - \text{prox}_{\gamma f_i}(x_k) \right) \right\rangle}_{:= P_1}.
\end{aligned}$$

Now let us focus on $P_1$, we have the following upper bound,

$$P_1 \leq \frac{1}{2}\|\gamma\nabla M^\gamma(x_k)\|^2 + \frac{1}{2}\left\|\frac{1}{n}\sum_{i=1}^n\left(\tilde{x}_{i,k+1} - \mathrm{prox}_{\gamma f_i}(x_k)\right)\right\|^2$$

$$\overset{(10)}{\leq} \frac{1}{2}\|\gamma\nabla M^\gamma(x_k)\|^2 + \frac{\varepsilon_1}{2}.$$

As a result, we have

$$\langle\gamma\nabla M^\gamma(x_k), g(x_k)\rangle \geq \frac{1}{2}\|\gamma\nabla M^\gamma(x_k)\| - \frac{\varepsilon_1}{2},$$

which holds for arbitrary $x_k$. This suggests that $b = \frac{1}{2}, c = \frac{\varepsilon_1}{2}$. On the other hand,

$$\mathbb{E}\left[\|g(x_k)\|^2\right] = \left\|\gamma\nabla M^\gamma(x_k) + \frac{1}{n}\sum_{i=1}^n\left(\tilde{x}_{i,k+1} - \mathrm{prox}_{\gamma f_i}(x_k)\right)\right\|^2$$

$$\overset{(14)}{\leq} 2\|\gamma\nabla M^\gamma(x_k)\|^2 + 2\left\|\frac{1}{n}\sum_{i=1}^n\left(\tilde{x}_{i,k+1} - \mathrm{prox}_{\gamma f_i}(x_k)\right)\right\|^2$$

$$\overset{(10)}{\leq} 2\|\gamma\nabla M^\gamma(x_k)\|^2 + 2\varepsilon_1.$$

Thus, we can choose $A = 0, B = 2, C = 2\varepsilon_1$. Since we have assumed Assumption 3 (Individual convexity) and Assumption 4 (Smoothness), it is easy to see that $M^\gamma$ is smooth, and we denote its smoothness constant as $L_\gamma$. It is therefore straightforward to see that our global objective $\gamma M^\gamma$ is $\gamma L_\gamma$-smooth. We also assume $f$ is $\mu$-strongly convex, which by Fact 8 indicates that $M^\gamma$ is $\frac{\mu}{1+\gamma L_{\max}}$ star strongly convex. We immediately obtain using Lemma 1 that $\gamma M^\gamma$ is $\frac{\gamma\mu}{4(1+\gamma L_{\max})}$-PL. Now, we have validated all the assumptions for using Theorem 6. Applying Theorem 6, we obtain that when the extrapolation parameter satisfies

$$0 < \alpha < \frac{1}{4}\cdot\min\left\{\frac{1}{\gamma L_\gamma}, \frac{2(1+\gamma L_{\max})}{\gamma\mu}\right\},$$

the last iterate $x_K$ of Algorithm 1 with each proximal operator solved inexactly according to Definition 1 satisfies

$$\mathcal{E}_K \leq \left(1 - \frac{\alpha\gamma\mu}{8(1+\gamma L_{\max})}\right)^K\mathcal{E}_0 + \frac{8\varepsilon_1\alpha L_\gamma(1+\gamma L_{\max})}{\mu} + \frac{4\varepsilon_1(1+\gamma L_{\max})}{\gamma\mu},$$

where $\mathcal{E}_k = \gamma M^\gamma(x_k) - M_{\inf}^\gamma$. Let us now prove that

$$\frac{1}{\gamma L_\gamma} < \frac{2(1+\gamma L_{\max})}{\gamma\mu}.$$

This is equivalent to prove

$$\mu < 2L_\gamma(1+\gamma L_{\max}),$$

which is always true since (27) holds. As a result, we can simplify the range of the extrapolation parameter to

$$0 < \alpha \leq \frac{1}{4\gamma L_\gamma}.$$

If we pick the largest possible $\alpha$, we have

$$\mathcal{E}_K \leq \left(1 - \frac{\mu}{32L_\gamma(1+\gamma L_{\max})}\right)^K\mathcal{E}_0 + \frac{6\varepsilon_1(1+\gamma L_{\max})}{\gamma\mu}.$$

This result is not directly comparable to that of Li et al. (2024a). However, using smoothness of $\gamma L_\gamma$, if we denote $\Delta_k = \|x_k - x_\star\|^2$ where $x_\star$ is a minimizer of both $M^\gamma$ and $f$ since we assume we are in the interpolation regime (Assumption 2), we have

$$\mathcal{E}_0 \leq \frac{\gamma L_\gamma}{2}\Delta_0.$$

Using star strong convexity, we have

$$\mathcal{E}_K \geq \frac{\gamma\mu}{2\left(1 + \gamma L_{\max}\right)} \Delta_K.$$

As a result, we can transform the above convergence guarantee into

$$\Delta_K \leq \left(1 - \frac{\mu}{32 L_\gamma\left(1 + \gamma L_{\max}\right)}\right)^K \frac{L_\gamma\left(1 + \gamma L_{\max}\right)}{\mu} \cdot \Delta_0 + 12\varepsilon_1 \cdot \left(\frac{1/\gamma + L_{\max}}{\mu}\right)^2.$$

This completes the proof.

### G.3   Proof of Theorem 2

Since we based our analysis on the theory of biased SGD, we first verify the validity of Assumption 6.

**Finding $b$ and $c$:**   Let us start with finding a lower bound on $\langle \gamma\nabla M^\gamma\left(x_k\right), \mathbb{E}\left[g(x_k)\right]\rangle$. We have

$$\langle \gamma M^\gamma\left(x_k\right), \mathbb{E}\left[g(x_k)\right]\rangle = \left\langle \gamma M^\gamma\left(x_k\right), \gamma M^\gamma\left(x_k\right) - \frac{1}{n}\sum_{i=1}^n \left(\tilde{x}_{i,k+1} - \text{prox}_{\gamma f_i}\left(x_k\right)\right)\right\rangle$$

$$= \left\|\gamma M^\gamma\left(x_k\right)\right\|^2 - \left\langle \gamma M^\gamma\left(x_k\right), \frac{1}{n}\sum_{i=1}^n \left(\tilde{x}_{i,k+1} - \text{prox}_{\gamma f_i}\left(x_k\right)\right)\right\rangle$$

$$\geq \left\|\gamma M^\gamma\left(x_k\right)\right\|^2 - \left\|\gamma M^\gamma\left(x_k\right)\right\| \cdot \left\|\frac{1}{n}\sum_{i=1}^n \left(\tilde{x}_{i,k+1} - \text{prox}_{\gamma f_i}\left(x_k\right)\right)\right\|,$$

where the last inequality is obtained using Cauchy-Schwarz inequality. We then utilize the convexity of $\|\cdot\|$ and obtain,

$$\langle \gamma M^\gamma\left(x_k\right), \mathbb{E}\left[g(x_k)\right]\rangle \;\; \geq \;\; \left\|\gamma M^\gamma\left(x_k\right)\right\|^2 - \left\|\gamma M^\gamma\left(x_k\right)\right\| \cdot \frac{1}{n}\sum_{i=1}^n \left\|\left(\tilde{x}_{i,k+1} - \text{prox}_{\gamma f_i}\left(x_k\right)\right)\right\|$$

$$\overset{(13)}{\geq} \;\; \left\|\gamma M^\gamma\left(x_k\right)\right\|^2 - \sqrt{\varepsilon_2}\left\|\gamma M^\gamma\left(x_k\right)\right\| \cdot \frac{1}{n}\sum_{i=1}^n \left\|x_k - \text{prox}_{\gamma f_i}\left(x_k\right)\right\|$$

$$= \;\; \left\|\gamma M^\gamma\left(x_k\right)\right\|^2 - \sqrt{\varepsilon_2}\left\|\gamma M^\gamma\left(x_k\right)\right\| \cdot \frac{1}{n}\sum_{i=1}^n \left\|\gamma\nabla M_{f_i}^\gamma\left(x_k\right)\right\|.$$

Notice that

$$\left\|\gamma\nabla M_{f_i}^\gamma\left(x_k\right)\right\| = \left\|\gamma\nabla M_{f_i}^\gamma\left(x_k\right) - \gamma\nabla M_{f_i}^\gamma\left(x_\star\right)\right\|,$$

holds for any $x_\star$ that is a minimizer of $M^\gamma\left(x\right)$ due to interpolation regime assumption. As a result, we can provide an upper bound based on smoothness of each individual $\gamma M_{f_i}^\gamma\left(x\right)$ using Fact 2,

$$\left\|\gamma\nabla M_{f_i}^\gamma\left(x_k\right) - \gamma\nabla M_{f_i}^\gamma\left(x_\star\right)\right\| \leq \frac{\gamma L_i}{1 + \gamma L_i}\left\|x_k - x_\star\right\|. \tag{28}$$

Thus,

$$\frac{1}{n}\sum_{i=1}^n \left\|\gamma\nabla M_{f_i}^\gamma\left(x_k\right)\right\| \leq \frac{1}{n}\sum_{i=1}^n \frac{\gamma L_i}{1 + \gamma L_i}\left\|x_k - x_\star\right\| \leq \frac{\gamma L_{\max}}{1 + \gamma L_{\max}} \cdot \left\|x_k - x_\star\right\|.$$

In addition, we have due to Cauchy-Schwarz inequality and the convexity of $M^\gamma\left(x\right)$

$$\left\|\nabla M^\gamma\left(x_k\right)\right\| \cdot \left\|x_k - x_\star\right\| \geq \langle\nabla M^\gamma\left(x_k\right), x_k - x_\star\rangle \geq M^\gamma\left(x_k\right) - M_{\inf}^\gamma, \tag{29}$$

and due to quadratic growth condition that

$$M^\gamma\left(x_k\right) - M_{\inf}^\gamma \geq \frac{\mu}{1 + \gamma L_{\max}} \cdot \frac{1}{2}\left\|x_k - x_\star\right\|^2. \tag{30}$$

Combining (29) and (30), we have

$$\frac{\mu}{2\left(1+\gamma L_{\max}\right)} \cdot \|x_k - x_\star\|^2 \overset{(29)+(30)}{\leq} \|\nabla M^\gamma\left(x_k\right)\| \cdot \|x_k - x_\star\|.$$

This indicates that

$$\|x_k - x_\star\| \leq \frac{2\left(1+\gamma L_{\max}\right)}{\mu} \|\nabla M^\gamma\left(x_k\right)\|. \tag{31}$$

Combining (28) and (31), we generate the following lower bound

$$\langle \gamma M^\gamma\left(x_k\right), \mathbb{E}\left[g(x_k)\right] \rangle \overset{(28)}{\geq} \|\gamma M^\gamma\left(x_k\right)\|^2 - \sqrt{\varepsilon_2} \|\gamma M^\gamma\left(x_k\right)\| \cdot \frac{\gamma L_{\max}}{1+\gamma L_{\max}} \|x_k - x_\star\|$$

$$\overset{(31)}{\geq} \|\gamma M^\gamma\left(x_k\right)\|^2 - \sqrt{\varepsilon_2} \cdot \frac{L_{\max}}{1+\gamma L_{\max}} \cdot \frac{2\left(1+\gamma L_{\max}\right)}{\mu} \|\gamma M^\gamma\left(x_k\right)\|^2$$

$$= \left(1 - \sqrt{\varepsilon_2} \cdot \frac{2L_{\max}}{\mu}\right) \cdot \|\gamma M^\gamma\left(x_k\right)\|^2.$$

Thus, as long as $\varepsilon_2 < \frac{\mu^2}{4L_{\max}^2}$, we have $b = 1 - \sqrt{\varepsilon_2} \cdot \frac{2L_{\max}}{\mu}$, and $c = 0$.

**Finding $A, B$ and $C$:** We start with expanding $\|g(x_k)\|^2$,

$$\mathbb{E}\left[\|g(x_k)\|^2\right] = \left\|\gamma M^\gamma\left(x_k\right) - \frac{1}{n}\sum_{i=1}^n \left(\tilde{x}_{i,k+1} - \mathrm{prox}_{\gamma f_i}\left(x_k\right)\right)\right\|^2$$

$$= \|\gamma M^\gamma\left(x_k\right)\|^2 + \underbrace{\left\|\frac{1}{n}\sum_{i=1}^n \left(\tilde{x}_{i,k+1} - \mathrm{prox}_{\gamma f_i}\left(x_k\right)\right)\right\|^2}_{:=T_2}$$

$$\underbrace{-2\left\langle \gamma M^\gamma\left(x_k\right), \frac{1}{n}\sum_{i=1}^n \left(\tilde{x}_{i,k+1} - \mathrm{prox}_{\gamma f_i}\left(x_k\right)\right)\right\rangle}_{:=T_3}. \tag{32}$$

It is easy to bound $T_2$ utilizing the convexity of $\|\cdot\|^2$,

$$T_2 \leq \frac{1}{n}\sum_{i=1}^n \left\|\tilde{x}_{i,k+1} - \mathrm{prox}_{\gamma f_i}\left(x_k\right)\right\|^2$$

$$\overset{(13)}{\leq} \frac{\varepsilon_2}{n}\sum_{i=1}^n \left\|x_k - \mathrm{prox}_{\gamma f_i}\left(x_k\right)\right\|^2 = \frac{\varepsilon_2}{n}\sum_{i=1}^n \left\|\gamma M_{f_i}^\gamma\left(x_k\right)\right\|^2.$$

Let $x_\star$ be a minimizer of $M^\gamma$, since we assume Assumption 2 holds, it is also a minimizer of each $M_{f_i}^\gamma$. As a result,

$$T_2 \leq \frac{\varepsilon_2}{n}\sum_{i=1}^n \left\|\gamma M_{f_i}^\gamma\left(x_k\right) - \gamma M_{f_i}^\gamma\left(x_\star\right)\right\|^2$$

$$\leq \frac{\varepsilon_2}{n}\sum_{i=1}^n \frac{2\gamma L_i}{1+\gamma L_i}\left(\gamma M_{f_i}^\gamma\left(x_k\right) - \gamma M_{f_i}^\gamma\left(x_\star\right)\right) \leq \frac{2\varepsilon_2\gamma L_{\max}}{1+\gamma L_{\max}} \cdot \left(\gamma M^\gamma\left(x_k\right) - \gamma M_{\inf}^\gamma\right). \tag{33}$$

We then consider $T_3$, and start with applying Cauchy-Schwarz inequality

$$T_3 \leq 2\|\gamma \nabla M^\gamma\left(x_k\right)\| \left\|\frac{1}{n}\sum_{i=1}^n \left(\tilde{x}_{i,k+1} - \mathrm{prox}_{\gamma f_i}\left(x_k\right)\right)\right\|. \tag{34}$$

Using the convexity of $\|\cdot\|$, we have

$$
\begin{aligned}
\left\| \frac{1}{n} \sum_{i=1}^n \left( \tilde{x}_{i,k+1} - \mathrm{prox}_{\gamma f_i}\left(x_k\right) \right) \right\| &\leq \frac{1}{n} \sum_{i=1}^n \left\| \tilde{x}_{i,k+1} - \mathrm{prox}_{\gamma f_i}\left(x_k\right) \right\| \\
&\overset{(13)}{\leq} \frac{\sqrt{\varepsilon_2}}{n} \sum_{i=1}^n \left\| x_k - \mathrm{prox}_{\gamma f_i}\left(x_k\right) \right\| \\
&\overset{(4)}{=} \frac{\sqrt{\varepsilon_2}}{n} \sum_{i=1}^n \left\| \gamma \nabla M_{f_i}^{\gamma}\left(x_k\right) - \gamma \nabla M_{f_i}^{\gamma}\left(x_\star\right) \right\| \\
&\overset{\text{Fact 2}}{\leq} \frac{\sqrt{\varepsilon_2}}{n} \sum_{i=1}^n \frac{\gamma L_i}{1 + \gamma L_i} \left\| x_k - x_\star \right\| \\
&\leq \frac{\sqrt{\varepsilon_2} \gamma L_{\max}}{1 + \gamma L_{\max}} \cdot \left\| x_k - x_\star \right\|.
\end{aligned}
$$

Utilizing (31), we have

$$
\begin{aligned}
\left\| \frac{1}{n} \sum_{i=1}^n \left( \tilde{x}_{i,k+1} - \mathrm{prox}_{\gamma f_i}\left(x_k\right) \right) \right\| &\leq \frac{\sqrt{\varepsilon_2}\gamma L_{\max}}{1+\gamma L_{\max}} \cdot \frac{2\left(1+\gamma L_{\max}\right)}{\mu} \left\| \nabla M^{\gamma}\left(x_k\right) \right\| \\
&= \frac{2\sqrt{\varepsilon_2} L_{\max}}{\mu} \cdot \left\| \gamma \nabla M^{\gamma}\left(x_k\right) \right\|
\end{aligned}
\tag{35}
$$

Plug the above inequality into (34), we have

$$
T_3 \leq \frac{4\sqrt{\varepsilon_2} L_{\max}}{\mu} \cdot \left\| \gamma \nabla M^{\gamma}\left(x_k\right) \right\|^2.
\tag{36}
$$

Combining (36) and (33), plug them into (32), we have

$$
\mathbb{E}\left[ \left\| g\left(x_k\right) \right\|^2 \right] \leq \frac{2\varepsilon_2 \gamma L_{\max}}{1+\gamma L_{\max}} \cdot \left( \gamma M^{\gamma}\left(x_k\right) - \gamma M_{\inf}^{\gamma} \right) + \left( 1 + \frac{4\sqrt{\varepsilon_2} L_{\max}}{\mu} \right) \cdot \left\| \gamma \nabla M^{\gamma}\left(x_k\right) \right\|^2.
$$

Thus, we have

$$
A = \frac{\varepsilon_2 \gamma L_{\max}}{1+\gamma L_{\max}}, \quad B = \frac{\mu + 4\sqrt{\varepsilon_2} L_{\max}}{\mu}, \quad C = 0.
$$

**Applying Theorem 6:** First, we list our the values appeared respectively,

$$
A = \frac{\varepsilon_2 \gamma L_{\max}}{1+\gamma L_{\max}}, \quad B = \frac{\mu + 4\sqrt{\varepsilon_2} L_{\max}}{\mu}, \quad b = \frac{\mu - 2\sqrt{\varepsilon_2} L_{\max}}{\mu},
$$

$$
C = c = 0.
$$

We know that the PL constant of $\gamma M^{\gamma}$ is given by $\frac{\gamma \mu}{4(1+\gamma L_{\max})}$ and the corresponding smoothness constant is $\gamma L_{\gamma}$. Applying Theorem 6, the range of $\alpha$ is given by

$$
0 < \alpha < \min \left\{ \underbrace{\frac{1}{\gamma L_{\gamma}} \cdot \frac{\mu - 2\sqrt{\varepsilon_2} L_{\max}}{\mu + 4\sqrt{\varepsilon_2} L_{\max} + 4\varepsilon_2 L_{\max}}}_{:=B_1}, \underbrace{\frac{4\left(1+\gamma L_{\max}\right)}{\gamma\left(\mu - 2\sqrt{\varepsilon_2} L_{\max}\right)}}_{:=B_2} \right\}.
\tag{37}
$$

Now notice that actually we can prove that for $\varepsilon_2 < \frac{\mu^2}{4L_{\max}^2}$, we have $B_2 > B_1$, and we can simplify the range of $\alpha$ to

$$
0 < \alpha \leq \frac{1}{\gamma L_{\gamma}} \cdot \frac{\mu - 2\sqrt{\varepsilon_2} L_{\max}}{\mu + 4\sqrt{\varepsilon_2} L_{\max} + 4\varepsilon_2 L_{\max}}.
$$

**Proof of $B_2 > B_1$** : It is easy to verify that the above inequality ($B_2 > B_1$) can be equivalently written as

$$4L_\gamma \left(1 + \gamma L_{\max}\right) \left(\mu + 4\sqrt{\varepsilon_2} L_{\max} + 4\varepsilon_2 L_{\max}\right) > \left(\mu - 2\sqrt{\varepsilon_2} L_{\max}\right)^2,$$

since when $\sqrt{\varepsilon_2} < \frac{\mu}{2L_{\max}}$, we have $\mu - 2\sqrt{\varepsilon_2} L_{\max} > 0$. We expand the right-hand side and obtain:

$$\left(\mu - 2\sqrt{\varepsilon_2} L_{\max}\right)^2 = \mu^2 - 4\sqrt{\varepsilon_2} L_{\max} + 4\varepsilon_2 L_{\max}^2 < 2\mu^2 - 4\sqrt{\varepsilon_2} L_{\max} < 2\mu^2.$$

For the left-hand side, as we have already shown in 27, we have

$$4L_\gamma \left(1 + \gamma L_{\max}\right) \left(\mu + 4\sqrt{\varepsilon_2} L_{\max} + 4\varepsilon_2 L_{\max}\right) \geq 4\mu \left(\mu + 4\sqrt{\varepsilon_2} L_{\max} + 2\varepsilon_2 L_{\max}\right) > 4\mu^2.$$

Combining the above inequality we arrive at $B_2 > B_1$.

**The convergence guarantee** : Given that we select $\alpha$ properly, we have

$$\mathcal{E}_K \leq \left(1 - \alpha \cdot \frac{\gamma \left(\mu - 2\sqrt{\varepsilon_2} L_{\max}\right)}{4 \left(1 + \gamma L_{\max}\right)}\right)^K \mathcal{E}_0,$$

where $\mathcal{E}_k = \gamma M^\gamma \left(x_k\right) - \gamma M_{\inf}^\gamma$. We do not have expectation here since we are in the full batch case. Specifically, if we choose the largest $\alpha$ possible, we have

$$\mathcal{E}_K \leq \left(1 - \frac{\mu}{4L_\gamma \left(1 + \gamma L_{\max}\right)} \cdot S\left(\varepsilon_2\right)\right)^k \mathcal{E}_0,$$

where

$$S(\varepsilon_2) = \frac{\left(\mu - 2\sqrt{\varepsilon_2} L_{\max}\right) \left(1 - 2\sqrt{\varepsilon_2} \frac{L_{\max}}{\mu}\right)}{\mu + 4\sqrt{\varepsilon_2} L_{\max} + 4\varepsilon_2 L_{\max}},$$

satisfies $0 < S(\varepsilon_2) \leq 1$ is the factor of slowing down due to inexact proximity operator evaluation. Using smoothness of $\gamma L_\gamma$, if we denote $\Delta_k = \|x_k - x_\star\|^2$ where $x_\star$ is a minimizer of both $M^\gamma$ and $f$ since we assume we are in the interpolation regime (Assumption 2), we have

$$\mathcal{E}_0 \leq \frac{\gamma L_\gamma}{2} \Delta_0.$$

Using star strong convexity (quadratic growth property), we have

$$\mathcal{E}_K \geq \frac{\gamma \mu}{2 \left(1 + \gamma L_{\max}\right)} \Delta_K.$$

As a result, we can transform the above convergence guarantee into

$$\Delta_K \leq \left(1 - \frac{\mu}{4L_\gamma \left(1 + \gamma L_{\max}\right)} \cdot S\left(\varepsilon_2\right)\right)^K \cdot \frac{L\gamma \left(1 + \gamma L_{\max}\right)}{\mu} \Delta_0.$$

This completes the proof.

### G.4 PROOF OF THEOREM 3

We start with formalizing the problem. Using (11) and (12), we can write the update rule of Algorithm 1 as

$$x_{k+1} = x_k - \alpha \cdot \left(\frac{1}{n} \sum_{i=1}^n \gamma \nabla M_{f_i}^\gamma \left(x_k\right) - \frac{1}{n} \sum_{i=1}^n \left(\tilde{x}_{i,k+1} - \text{prox}_{\gamma f_i} \left(x_k\right)\right)\right). \tag{38}$$

Since by Definition 4, we have $\left\|\tilde{x}_{i,k+1} - \text{prox}_{\gamma f_i} \left(x_k\right)\right\|^2 \leq \varepsilon_2 \left\|\gamma \nabla M_{f_i}^\gamma \left(x_k\right)\right\|^2$, we can view the left hand side as a compressed version of the true gradient. Specifically, there are two possible perspectives:

(I). Let $\mathcal{C}_i(\cdot)$ be the compressing mapping with the $i$-th client, $i \in \{1, 2, \ldots, n\}$, defined as

$$\mathcal{C}_i\left(\gamma\nabla M_{f_i}^\gamma(x_k)\right) := \gamma\nabla M_{f_i}^\gamma(x_k) - \left(\tilde{x}_{i,k+1} - \text{prox}_{\gamma f_i}(x_k)\right).$$

In this way, we reformulate (38) as

$$x_{k+1} = x_k - \alpha \cdot \frac{1}{n}\sum_{i=1}^n \mathcal{C}_i\left(\gamma\nabla M_{f_i}^\gamma(x_k)\right). \tag{39}$$

(39) is exactly DCGD with biased compression. We can easily prove that

$$\mathcal{C}_i \in \mathbb{B}^1\left(\alpha = 1 - 2\sqrt{\varepsilon_2}, \beta = \frac{1 - \sqrt{\varepsilon_2}}{1 + \varepsilon_2}\right)$$

$$\mathcal{C}_i \in \mathbb{B}^2\left(\xi = 1 - \sqrt{\varepsilon_2}, \beta = \frac{1 - \sqrt{\varepsilon_2}}{1 + \varepsilon_2}\right)$$

$$\mathcal{C}_i \in \mathbb{B}^3\left(\delta = \frac{1}{1 - \varepsilon_2}\right).$$

However, DCGD with biased compression may fail to converge even if the above formulation of compression mapping seems quite nice. For an example of such failure, we refer the readers to Beznosikov et al. (2023, Example 1). This limitation can be circumvented by employing an error feedback mechanism; however, this approach requires modifications to the original algorithm. We therefore leave it as a future research direction.

(II). We can also view it as if we are in the single node case. Let $\mathcal{C}(\cdot)$ be the compressing mapping defined as

$$\mathcal{C}\left(\nabla M^\gamma(x_k)\right) := \frac{1}{n}\sum_{i=1}^n \gamma\nabla M_{f_i}^\gamma(x_k) - \frac{1}{n}\sum_{i=1}^n\left(\tilde{x}_{i,k+1} - \text{prox}_{\gamma f_i}(x_k)\right)$$

$$= \gamma\nabla M^\gamma(x_k) - \frac{1}{n}\sum_{i=1}^n\left(\tilde{x}_{i,k+1} - \text{prox}_{\gamma f_i}(x_k)\right). \tag{40}$$

This formulation leads us to the convergence guarantee appeared in Theorem 3, as we illustrate below.

Let us first analyze $\mathcal{C}$ defined in (40). We will verify it belongs to $\mathbb{B}^3(\delta)$. The inequality we want to prove can be written equivalently as

$$\left\|\gamma\nabla M^\gamma(x_k) - \frac{1}{n}\sum_{i=1}^n\left(\tilde{x}_{i,k+1} - \text{prox}_{\gamma f_i}(x_k)\right) - \gamma\nabla M^\gamma(x_k)\right\|^2 \leq \left(1 - \frac{1}{\delta}\right)\|\gamma\nabla M^\gamma(x_k)\|^2, \tag{41}$$

which is exactly

$$\left\|\frac{1}{n}\sum_{i=1}^n\left(\tilde{x}_{i,k+1} - \text{prox}_{\gamma f_i}(x_k)\right)\right\|^2 \leq \|\gamma\nabla M^\gamma(x_k)\|^2$$

For the left-hand side, using the convexity of $\|\cdot\|^2$ in combination with Definition 4, we obtain

$$\left\|\frac{1}{n}\sum_{i=1}^n\left(\tilde{x}_{i,k+1} - \text{prox}_{\gamma f_i}(x_k)\right)\right\|^2 \leq \frac{1}{n}\sum_{i=1}^n\left\|\tilde{x}_{i,k+1} - \text{prox}_{\gamma f_i}(x_k)\right\|^2$$

$$\leq \frac{\varepsilon_2}{n}\sum_{i=1}^n\left\|x_k - \text{prox}_{\gamma f_i}(x_k)\right\|^2.$$

Let $x_\star$ be a minimizer of $f$, since we assume Assumption 2 holds, by Fact 7, it is also a minimizer of $\gamma M^\gamma$,

$$\frac{\varepsilon_2}{n}\sum_{i=1}^{n}\left\|x_k - \mathrm{prox}_{\gamma f_i}(x_k)\right\|^2 \overset{(4)}{=} \frac{\varepsilon_2}{n}\sum_{i=1}^{n}\left\|\gamma\nabla M_{f_i}^\gamma(x_k)\right\|^2$$

$$= \frac{\varepsilon_2}{n}\sum_{i=1}^{n}\left\|\gamma\nabla M_{f_i}^\gamma(x_k) - \gamma\nabla M_{f_i}^\gamma(x_\star)\right\|^2$$

$$\overset{\text{Fact 2}}{\leq} \frac{2\varepsilon_2}{n}\sum_{i=1}^{n}\frac{\gamma L_i}{1+\gamma L_i}\left(\gamma M_{f_i}^\gamma(x_k) - \gamma M_{f_i}^\gamma(x_\star)\right)$$

$$\leq \frac{2\varepsilon_2\gamma L_{\max}}{1+\gamma L_{\max}}\left(\gamma M^\gamma(x_k) - \gamma M^\gamma(x_\star)\right).$$

We then notice that as it is illustrated by Lemma 1, we have

$$\left(1-\frac{1}{\delta}\right)\|\gamma\nabla M^\gamma(x_k)\|^2 \geq \left(1-\frac{1}{\delta}\right)\frac{\gamma\mu}{2(1+\gamma L_{\max})}\left(\gamma M^\gamma(x_k) - \gamma M^\gamma(x_\star)\right).$$

Combining the above two inequalities, we know that the following inequality is a sufficient condition for (41),

$$\frac{2\varepsilon_2\gamma L_{\max}}{1+\gamma L_{\max}}\left(\gamma M^\gamma(x_k) - \gamma M^\gamma(x_\star)\right) \leq \left(1-\frac{1}{\delta}\right)\frac{\gamma\mu}{2(1+\gamma L_{\max})}\left(\gamma M^\gamma(x_k) - \gamma M^\gamma(x_\star)\right).$$

It is easy to check that if we pick

$$\delta = \frac{\mu}{\mu - 4\varepsilon_2 L_{\max}} > 0, \tag{42}$$

the condition is met. However, for this to hold, we must ensure that $\varepsilon_2 < \frac{\mu}{4L_{\max}}$.

As we mentioned in Appendix D, Beznosikov et al. (2023) provided the theory of CGD with biased compressor belongs to $\mathbb{B}^3(\delta)$. We have already shown that $\mathcal{C} \in \mathbb{B}^3\left(\delta = \frac{\mu}{\mu-4\varepsilon_2 L_{\max}}\right)$, when $\varepsilon_2 < \frac{4L_{\max}}{\mu}$. Notice that our objective $\gamma M^\gamma$ is $\gamma L_\gamma$-smooth and $\frac{\gamma\mu}{1+\gamma L_{\max}}$-PL.[5] Therefore, as long as $0 < \alpha \leq \frac{1}{\gamma L_\gamma}$ and $\varepsilon_2 < \frac{\mu}{4L_{\max}}$, we have

$$\mathcal{E}_K \leq \left(1 - \frac{\mu - 4\varepsilon_2 L_{\max}}{\mu}\cdot\frac{\gamma\mu}{4(1+\gamma L_{\max})}\cdot\alpha\right)^K \mathcal{E}_0,$$

Taking $\alpha = \frac{1}{\gamma L_\gamma}$, which is the largest step size possible, we can further simplify the above convergence into

$$M^\gamma(x_k) - M_\star^\gamma \leq \left(1 - \left(1 - \frac{4\varepsilon_2 L_{\max}}{\mu}\right)\cdot\frac{\mu}{4L_\gamma(1+\gamma L_{\max})}\right)^K \left(M^\gamma(x_0) - M^{\gamma\star}\right).$$

Using smoothness of $\gamma L_\gamma$, if we denote $\Delta_k = \|x_k - x_\star\|^2$ where $x_\star$ is a minimizer of both $M^\gamma$ and $f$ since we assume we are in the interpolation regime (Assumption 2), we have

$$\mathcal{E}_0 \leq \frac{\gamma L_\gamma}{2}\Delta_0.$$

Using star strong convexity (quadratic growth property), we have

$$\mathcal{E}_K \geq \frac{\gamma\mu}{2(1+\gamma L_{\max})}\Delta_K.$$

As a result, we can transform the above convergence guarantee into

$$\Delta_K \leq \left(1 - \left(1 - \frac{4\varepsilon_2 L_{\max}}{\mu}\right)\cdot\frac{\mu}{4L_\gamma(1+\gamma L_{\max})}\right)^K \cdot\frac{L_\gamma(1+\gamma L_{\max})}{\mu}\Delta_0.$$

This completes the proof.

---

[5]Theorem 7 remains valid if we replace $f$ being strongly convex with PL.

### G.5 PROOF OF THEOREM 4

Notice that we assume each $f_i$ is $L_i$-smooth and convex. The local optimization of each client can be written as

$$\min_{z \in \mathbb{R}^d} \left\{ A_{k,i}^{\gamma}(z) = f_i(z) + \frac{1}{2\gamma} \|z - x_k\|^2 \right\},$$

It is easy to see that $A_{k,i}^{\gamma}(z)$ is $L_i + \frac{1}{\gamma}$-smooth and $\frac{1}{\gamma}$-strongly convex. We first provide the convergence theory of GD for reference.

**Theory of GD:** For a $\widehat{\mu}$-strongly convex, $\widehat{L}$-smooth function $\phi$, the algorithm can be formulated as

$$z_{t+1} = z_t - \eta \nabla \phi(z_t), \tag{GD}$$

where $z_t$ is the iterate in the $t$-th iteration, and $\eta > 0$ is the step size. GD with step size $\eta \in (0, \frac{1}{\widehat{L}}]$ generates iterates that satisfy

$$\|z_t - z_\star\|^2 \le (1 - \eta\widehat{\mu})^t \|z_0 - z_\star\|^2,$$

where $z_\star$ is a minimizer of $\phi$, $t$ is the number of iterations (number of gradient evaluations).

**Approximation satisfying Definition 3:** Notice that $\text{prox}_{\gamma f_i}(x_k)$ is the minimizer of $A_{k,i}^{\gamma}(z)$ and $z_0 = x_k$. As a result, if we run GD with the largest step size $\frac{\gamma}{1+\gamma L_i}$,

$$\left\|z_t - \text{prox}_{\gamma f_i}(x_k)\right\|^2 \le \left(1 - \frac{1}{1 + \gamma L_i}\right)^t \left\|x_k - \text{prox}_{\gamma f_i}(x_k)\right\|^2 \tag{43}$$

We have

$$t = \mathcal{O}\left((1 + \gamma L_i) \log\left(\frac{\left\|x_k - \text{prox}_{\gamma f_i}(x_k)\right\|^2}{\varepsilon_1}\right)\right).$$

The unknown term $\left\|x_k - \text{prox}_{\gamma f_i}(x_k)\right\|^2$ within the log can be bounded by

$$\left\|x_k - \text{prox}_{\gamma f_i}(x_k)\right\|^2 = \|z_0 - z_\star\|^2$$
$$\le \gamma^2 \left\|\nabla A_{k,i}^{\gamma}(z_0) - \nabla A_{k,i}^{\gamma}(z_\star)\right\|^2 = \|\gamma \nabla f_i(x_k)\|^2, \tag{44}$$

which can be easily calculated.

**Approximation satisfying Definition 4:** According to (43), we have

$$t = \mathcal{O}\left((1 + \gamma L_i) \log\left(\frac{1}{\varepsilon_2}\right)\right).$$

This completes the proof.

### G.6 PROOF OF THEOREM 5

We first provide the theory of AGD (Nesterov, 2004).

**Theory of AGD:** For a $\widehat{\mu}$-strongly convex, $\widehat{L}$-smooth function $\phi$, the algorithm can be formulated as

$$y_{t+1} = z_t + \alpha(z_t - z_{t-1})$$
$$z_{t+1} = y_{t+1} - \eta \nabla \phi(y_{t+1}), \tag{AGD}$$

where $z_t, y_t$ are iterates, $\eta > 0$ is the step size, $\alpha > 0$ is the momentum parameter. AGD with step size $\eta = \frac{1}{\widehat{L}}$, momentum $\alpha = \frac{\sqrt{\widehat{L}} - \sqrt{\widehat{\mu}}}{\sqrt{\widehat{L}} + \sqrt{\widehat{\mu}}}$ generates iterates that satisfy

$$\|z_t - z_\star\|^2 \le \frac{2\widehat{L}}{\widehat{\mu}} \cdot \left(1 - \sqrt{\frac{\widehat{\mu}}{\widehat{L}}}\right)^t \|z_0 - z_\star\|^2,$$

where $z_\star$ is a minimizer of $\phi$, $t$ is the number of iterations (number of gradient evaluations).

**Approximation satisfying Definition 3:** Notice that $\mathrm{prox}_{\gamma f_i}(x_k)$ is the minimizer of $A_{k,i}^{\gamma}(z)$ and $z_0 = x_k$. As a result, if we run AGD with the step size $\frac{\gamma}{1+\gamma L_i}$ and momentum $\alpha = \frac{\sqrt{1+\gamma L_i}-1}{\sqrt{1+\gamma L_i}+1}$,

$$\left\| z_t - \mathrm{prox}_{\gamma f_i}(x_k) \right\|^2 \leq 2 \cdot (1 + \gamma L_i) \left( 1 - \frac{1}{\sqrt{1+\gamma L_i}} \right)^t \left\| x_k - \mathrm{prox}_{\gamma f_i}(x_k) \right\|^2. \tag{45}$$

We have

$$t = \mathcal{O}\left( \sqrt{1+\gamma L_i} \log \left( \frac{(1+\gamma L_i) \cdot \left\| x_k - \mathrm{prox}_{\gamma f_i}(x_k) \right\|^2}{\varepsilon_1} \right) \right)$$

Similar to the proof of Theorem 4, since we have according to (44),

$$\left\| x_k - \mathrm{prox}_{\gamma f_i}(x_k) \right\|^2 \leq \left\| \gamma \nabla f_i(x_k) \right\|^2,$$

it is straightforward to determine the number of local iterations needed.

**Approximation satisfying Definition 4:** Using (45), we have

$$t = \mathcal{O}\left( \sqrt{1+\gamma L_i} \log \left( \frac{1+\gamma L_i}{\varepsilon_2} \right) \right).$$

### G.7 PROOF OF THEOREM 8

In this case, the gradient estimator is defined as

$$g(x_k) = \frac{1}{\tau} \sum_{i \in S_k} \left( \gamma \nabla M_{f_i}^{\gamma}(x_k) - \left( \tilde{x}_{i,k+1} - \mathrm{prox}_{\gamma f_i}(x_k) \right) \right). \tag{46}$$

Notice that we have

$$\left\langle \gamma \nabla M^{\gamma}(x_k), \mathbb{E}\left[ g(x_k) \right] \right\rangle$$

$$= \left\langle \gamma \nabla M^{\gamma}(x_k), \mathbb{E}\left[ \frac{1}{\tau} \sum_{i \in S_k} \gamma \nabla M_{f_i}^{\gamma}(x_k) - \frac{1}{\tau} \sum_{i \in S_k} \left( \tilde{x}_{i,k+1} - \mathrm{prox}_{\gamma f_i}(x_k) \right) \right] \right\rangle$$

$$= \left\langle \gamma \nabla M^{\gamma}(x_k), \gamma \nabla M^{\gamma}(x_k) - \frac{1}{n} \sum_{i=1}^{n} \left( \tilde{x}_{i,k+1} - \mathrm{prox}_{\gamma f_i}(x_k) \right) \right\rangle.$$

Using the same technique in the proof of Theorem 2, we are able to obtain that

$$\left\langle \gamma \nabla M^{\gamma}(x_k), \mathbb{E}\left[ g(x_k) \right] \right\rangle \geq \left( 1 - \frac{2\sqrt{\varepsilon_2} L_{\max}}{\mu} \right) \cdot \left\| \gamma \nabla M^{\gamma}(x_k) \right\|^2.$$

Thus, as long as we pick $\varepsilon_2 < \frac{\mu^2}{4L_{\max}^2}$, we can pick $b = 1 - \sqrt{\varepsilon_2} \cdot \frac{2L_{\max}}{\mu}$ and $c = 0$. We then compute $\mathbb{E}\left[ \left\| g(x_k) \right\|^2 \right]$,

$$\mathbb{E}\left[ \left\| g(x_k) \right\|^2 \right] = \mathbb{E}\left[ \left\| \frac{1}{\tau} \sum_{i \in S_k} \gamma \nabla M_{f_i}^{\gamma}(x_k) - \frac{1}{\tau} \sum_{i \in S_k} \left( \tilde{x}_{i,k+1} - \mathrm{prox}_{\gamma f_i}(x_k) \right) \right\|^2 \right]$$

$$= \underbrace{\mathbb{E}\left[ \left\| \frac{1}{\tau} \sum_{i \in S_k} \gamma \nabla M_{f_i}^{\gamma}(x_k) \right\|^2 \right]}_{:=T_1} + \underbrace{\mathbb{E}\left[ \left\| \frac{1}{\tau} \sum_{i \in S_k} \left( \tilde{x}_{i,k+1} - \mathrm{prox}_{\gamma f_i}(x_k) \right) \right\|^2 \right]}_{:=T_2}$$

$$\underbrace{- 2\mathbb{E}\left[ \left\langle \frac{1}{\tau} \sum_{i \in S_k} \gamma \nabla M_{f_i}^{\gamma}(x_k), \frac{1}{\tau} \sum_{i \in S_k} \left( \tilde{x}_{i,k+1} - \mathrm{prox}_{\gamma f_i}(x_k) \right) \right\rangle \right]}_{:=T_3}.$$

We try to provide upper bounds for those terms separately.

**Term $T_1$:** We have

$$T_1 = \frac{n-\tau}{\tau(n-1)} \cdot \frac{1}{n} \sum_{i=1}^{n} \left\| \gamma \nabla M_{f_i}^{\gamma}(x_k) \right\|^2 + \frac{n(\tau-1)}{\tau(n-1)} \cdot \|\gamma \nabla M^{\gamma}(x_k)\|^2.$$

Using smoothness of $\gamma M_{f_i}^{\gamma}$ and the fact that we are in the interpolation regime, we have

$$\begin{aligned}
T_1 &= \frac{n-\tau}{\tau(n-1)} \cdot \frac{1}{n} \sum_{i=1}^{n} \left\| \gamma \nabla M_{f_i}^{\gamma}(x_k) - \gamma \nabla M_{f_i}^{\gamma}(x_\star) \right\|^2 + \frac{n(\tau-1)}{\tau(n-1)} \cdot \|\gamma \nabla M^{\gamma}(x_k)\|^2 \\
&\leq \frac{n-\tau}{\tau(n-1)} \cdot \frac{1}{n} \sum_{i=1}^{n} \frac{2\gamma L_i}{1+\gamma L_i} \cdot \left( \gamma M_{f_i}^{\gamma}(x_k) - \gamma \left( M_{f_i}^{\gamma} \right)_{\inf} \right) + \frac{n(\tau-1)}{\tau(n-1)} \cdot \|\gamma \nabla M^{\gamma}(x_k)\|^2 \\
&\leq \frac{n-\tau}{\tau(n-1)} \cdot \frac{2\gamma L_{\max}}{1+\gamma L_{\max}} \cdot \left( \gamma M^{\gamma}(x_k) - \gamma M_{\inf}^{\gamma} \right) + \frac{n(\tau-1)}{\tau(n-1)} \cdot \|\gamma \nabla M^{\gamma}(x_k)\|^2 . \quad (47)
\end{aligned}$$

**Term $T_2$:** It is easy to see that using convexity of the squared Euclidean norm, we have

$$\begin{aligned}
T_2 &\leq \mathbb{E} \left[ \frac{1}{\tau} \sum_{i \in S_k} \left\| \tilde{x}_{i,k+1} - \operatorname{prox}_{\gamma f_i}(x_k) \right\|^2 \right] \\
&= \frac{1}{n} \sum_{i=1}^{n} \left\| \tilde{x}_{i,k+1} - \operatorname{prox}_{\gamma f_i}(x_k) \right\|^2 \overset{(13)}{\leq} \frac{\varepsilon_2}{n} \sum_{i=1}^{n} \left\| \gamma \nabla M_{f_i}^{\gamma}(x_k) \right\|^2 .
\end{aligned}$$

Using smoothness of each individual $\gamma M_{f_i}^{\gamma}(x_k)$ and the fact we are in the interpolation regime, we have

$$T_2 \leq \frac{2\varepsilon_2 \gamma L_{\max}}{1+\gamma L_{\max}} \left( \gamma M^{\gamma}(x_k) - \gamma M_{\inf}^{\gamma} \right). \quad (48)$$

**Term $T_3$:** We have

$$\begin{aligned}
T_3 &= -2 \cdot \frac{n-\tau}{\tau(n-1)} \cdot \frac{1}{n} \sum_{i=1}^{n} \left\langle \gamma \nabla M_{f_i}^{\gamma}(x_k), \tilde{x}_{i,k+1} - \operatorname{prox}_{\gamma f_i}(x_k) \right\rangle \\
&\quad - 2 \cdot \frac{n(\tau-1)}{\tau(n-1)} \cdot \left\langle \gamma \nabla M^{\gamma}(x_k), \frac{1}{n} \sum_{i=1}^{n} \left( \tilde{x}_{i,k+1} - \operatorname{prox}_{\gamma f_i}(x_k) \right) \right\rangle .
\end{aligned}$$

Using Cauchy-Schwarz inequality and convexity, we further obtain

$$\begin{aligned}
T_3 &\leq 2 \cdot \frac{n-\tau}{\tau(n-1)} \cdot \frac{1}{n} \sum_{i=1}^{n} \left\| \gamma \nabla M_{f_i}^{\gamma}(x_k) \right\| \left\| \tilde{x}_{i,k+1} - \operatorname{prox}_{\gamma f_i}(x_k) \right\| \\
&\quad + 2 \cdot \frac{n(\tau-1)}{\tau(n-1)} \|\gamma \nabla M^{\gamma}(x_k)\| \cdot \frac{1}{n} \sum_{i=1}^{n} \left\| \tilde{x}_{i,k+1} - \operatorname{prox}_{\gamma f_i}(x_k) \right\| .
\end{aligned}$$

Using similar approaches in the previous paragraphs, we have

$T_3$

$$
\overset{(13)}{\leq} \frac{2\left(n-\tau\right)}{\tau\left(n-1\right)} \cdot \frac{\sqrt{\varepsilon_2}}{n} \sum_{i=1}^{n} \left\|\gamma \nabla M_{f_i}^{\gamma}\left(x_k\right)\right\|^2 + \frac{2n\left(\tau-1\right)}{\tau\left(n-1\right)} \left\|\gamma M^{\gamma}\left(x_k\right)\right\| \frac{\sqrt{\varepsilon_2}}{n} \cdot \sum_{i=1}^{n} \left\|\gamma \nabla M_{f_i}^{\gamma}\left(x_k\right)\right\|
$$

$$
\leq \frac{2\left(n-\tau\right)}{\tau\left(n-1\right)} \cdot \frac{\sqrt{\varepsilon_2}}{n} \sum_{i=1}^{n} \left\|\gamma \nabla M_{f_i}^{\gamma}\left(x_k\right) - \gamma \nabla M_{f_i}^{\gamma}\left(x_\star\right)\right\|^2
$$

$$
+ \frac{2n\left(\tau-1\right)}{\tau\left(n-1\right)} \left\|\gamma M^{\gamma}\left(x_k\right)\right\| \frac{\sqrt{\varepsilon_2}}{n} \cdot \sum_{i=1}^{n} \left\|\gamma \nabla M_{f_i}^{\gamma}\left(x_k\right) - \gamma \nabla M_{f_i}^{\gamma}\left(x_k\right)\right\|
$$

$$
\leq \frac{4\sqrt{\varepsilon_2}\left(n-\tau\right)}{\tau\left(n-1\right)} \cdot \frac{\gamma L_{\max}}{1+\gamma L_{\max}} \left(\gamma M^{\gamma}\left(x_k\right) - \gamma M_{\inf}^{\gamma}\right)
$$

$$
+ \frac{4\sqrt{\varepsilon_2}n\left(\tau-1\right)}{\tau\left(n-1\right)} \cdot \frac{\gamma L_{\max}}{1+\gamma L_{\max}} \left\|x_k - x_\star\right\| \left\|\gamma \nabla M^{\gamma}\left(x_k\right)\right\|
$$

$$
\overset{(25)}{\leq} \frac{4\sqrt{\varepsilon_2}\left(n-\tau\right)}{\tau\left(n-1\right)} \cdot \frac{\gamma L_{\max}}{1+\gamma L_{\max}} \left(\gamma M^{\gamma}\left(x_k\right) - \gamma M_{\inf}^{\gamma}\right)
$$

$$
+ \frac{4\sqrt{\varepsilon_2}n\left(\tau-1\right)}{\tau\left(n-1\right)} \cdot \frac{L_{\max}}{\mu} \left\|\gamma \nabla M^{\gamma}\left(x_k\right)\right\|^2 . \tag{49}
$$

Combining (47), (48) and (49), we have

$$
\sum_{i=1}^{3} T_i \leq 2\left(\varepsilon_2 + \frac{2\sqrt{\varepsilon_2}\left(n-\tau\right)}{\tau\left(n-1\right)} + \frac{\left(n-\tau\right)}{\tau\left(n-1\right)}\right) \cdot \frac{\gamma L_{\max}}{1+\gamma L_{\max}} \cdot \left(\gamma M^{\gamma}\left(x_k\right) - \gamma M_{\inf}^{\gamma}\right)
$$

$$
+ \left(\frac{n\left(\tau-1\right)}{\tau\left(n-1\right)} + \frac{4\sqrt{\varepsilon_2}n\left(\tau-1\right)}{\tau\left(n-1\right)}\right) \cdot \frac{L_{\max}}{\mu} \cdot \left\|\gamma M^{\gamma}\left(x_k\right)\right\|^2 . \tag{50}
$$

Therefore, it is easy to see that we can pick

$$
A = \left(\varepsilon_2 + \frac{2\sqrt{\varepsilon_2}\left(n-\tau\right)}{\tau\left(n-1\right)} + \frac{\left(n-\tau\right)}{\tau\left(n-1\right)}\right) \cdot \frac{\gamma L_{\max}}{1+\gamma L_{\max}}
$$

$$
B = \left(\frac{n\left(\tau-1\right)}{\tau\left(n-1\right)} + \frac{4\sqrt{\varepsilon_2}n\left(\tau-1\right)}{\tau\left(n-1\right)}\right) \cdot \frac{L_{\max}}{\mu}, \qquad C = 0.
$$

Applying Theorem 4 of Demidovich et al. (2024), we list the corresponding values of $A, B, C, b, c \geq 0$ below,

$$
A = \frac{\gamma L_{\max}}{1+\gamma L_{\max}} \left(\varepsilon_2 + \frac{2\sqrt{\varepsilon_2}\left(n-\tau\right)}{\tau\left(n-1\right)} + \frac{\left(n-\tau\right)}{\tau\left(n-1\right)}\right)
$$

$$
B = \frac{n\left(\tau-1\right)}{\tau\left(n-1\right)} \left(1 + \frac{4\sqrt{\varepsilon_2}L_{\max}}{\mu}\right), \quad C = 0
$$

$$
b = \frac{\mu - 2\sqrt{\varepsilon_2}L_{\max}}{\mu}, \quad c = 0.
$$

We know that the PL constant of $\gamma M^{\gamma}$ is given by $\frac{\gamma\mu}{4(1+\gamma L_{\max})}$ and the corresponding smoothness constant is $\gamma L_{\gamma}$. As a result, when $\alpha > 0$ satisfies

$$
\alpha < \underbrace{\frac{1}{\gamma L_{\gamma}} \cdot \frac{\mu - 2\sqrt{\varepsilon_2}L_{\max}}{\mu + 4\varepsilon_2 L_{\max} + 4\sqrt{\varepsilon_2}L_{\max} + \frac{n-\tau}{\tau(n-1)} \cdot \left(4L_{\max} + 4\sqrt{\varepsilon_2}L_{\max} - \mu\right)}}_{:=B_1'},
$$

and

$$
\alpha < \underbrace{\frac{4\left(1+\gamma L_{\max}\right)}{\gamma\left(\mu - 2\sqrt{\varepsilon_2}L_{\max}\right)}}_{=B_2},
$$

we can obtain a convergence guarantee for the algorithm. Notice that $B_1' \leq B_1 < B_2$[6], thus we can further simplify the range of $\alpha$ to

$$\alpha \leq \underbrace{\frac{1}{\gamma L_\gamma} \cdot \frac{\mu - 2\sqrt{\varepsilon_2} L_{\max}}{\mu + 4\varepsilon_2 L_{\max} + 4\sqrt{\varepsilon_2} L_{\max} + \frac{n-\tau}{\tau(n-1)} \cdot \left(4L_{\max} + 4\sqrt{\varepsilon_2} L_{\max} - \mu\right)}}_{:=B_1'}.$$

Given that we select $\alpha$ properly, we have

$$\mathbb{E}\left[\mathcal{E}_K\right] \leq \left(1 - \alpha \cdot \frac{\gamma\left(\mu - 2\sqrt{\varepsilon_2} L_{\max}\right)}{4\left(1 + \gamma L_{\max}\right)}\right)^K \mathcal{E}_0.$$

Specifically, if we choose the largest $\alpha$ possible, we have

$$\mathbb{E}\left[\mathcal{E}_K\right] \leq \left(1 - \frac{\mu}{4L_\gamma\left(1 + \gamma L_{\max}\right)} \cdot S\left(\varepsilon_2, \tau\right)\right)^K \mathcal{E}_0,$$

where $S\left(\varepsilon_2, \tau\right)$ is defined as

$$S\left(\varepsilon_2, \tau\right) = \frac{\left(\mu - 2\sqrt{\varepsilon_2} L_{\max}\right)\left(1 - 2\sqrt{\varepsilon_2}\frac{L_{\max}}{\mu}\right)}{\mu + 4\varepsilon_2 L_{\max} + 4\sqrt{\varepsilon_2} L_{\max} + \frac{n-\tau}{\tau(n-1)} \cdot \left(4L_{\max} + 4\sqrt{\varepsilon_2} L_{\max} - \mu\right)},$$

satisfying

$$0 < S\left(\varepsilon_2, \tau\right) \leq 1.$$

Using smoothness of $\gamma L_\gamma$, if we denote $\Delta_k = \|x_k - x_\star\|^2$ where $x_\star$ is a minimizer of both $M^\gamma$ and $f$ since we assume we are in the interpolation regime (Assumption 2), we have

$$\mathcal{E}_0 \leq \frac{\gamma L_\gamma}{2}\Delta_0.$$

Using star strong convexity (quadratic growth property), we have

$$\mathcal{E}_K \geq \frac{\gamma\mu}{2\left(1 + \gamma L_{\max}\right)}\Delta_K.$$

As a result, we can transform the above convergence guarantee into

$$\mathbb{E}\left[\Delta_K\right] \leq \left(1 - \frac{\mu}{4L_\gamma\left(1 + \gamma L_{\max}\right)} \cdot S\left(\varepsilon_2, \tau\right)\right)^K \cdot \frac{L\gamma\left(1 + \gamma L_{\max}\right)}{\mu}\Delta_0.$$

This completes the proof.

# H EXPERIMENTS

We describe the settings for the numerical experiments and the corresponding results to validate our theoretical findings. We are interested in the following optimization problem in the distributed setting,

$$\min_{x \in \mathbb{R}^d}\left\{f(x) = \frac{1}{n}\sum_{i=1}^{n} f_i\left(x\right)\right\}.$$

Here $n$ denotes the number of clients, $d$ is the dimension, each function $f_i : \mathbb{R}^d \mapsto \mathbb{R}$ has the following form

$$f_i(x) = \frac{1}{2}x^\top \boldsymbol{A}_i x + b_i^\top x + c_i,$$

where $\boldsymbol{A}_i \in \mathbb{S}_+^d, b_i \in \mathbb{R}^d, c_i \in \mathbb{R}$. Specifically, we pick $n = 20$ and $d = 300$ for the experiments. Notice that we have

$$\nabla f_i(x) = \boldsymbol{A}_i x - b_i; \qquad \nabla^2 f_i(x) = \boldsymbol{A}_i \succeq \boldsymbol{O}_d,$$

---

[6]The definition of $B_1$ is given in (37)

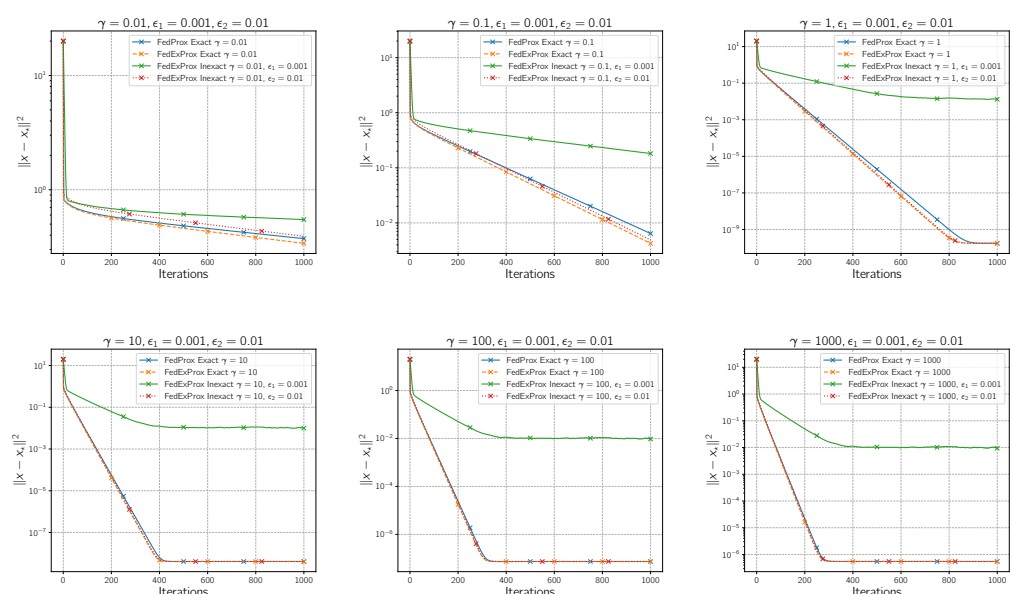

Figure 2: Comparison of FedProx, FedExProx with exact proximal evaluations, FedExProx with $\varepsilon_1$-absolute approximation and FedExProx with $\varepsilon_2$-relative approximation. In this case, we fix $\varepsilon_1 = 0.001$, $\varepsilon_2 = 0.01$ and pick the local step size $\gamma \in \{1000, 100, 10, 1, 0.1, 0.01\}$. The $y$-axis is the squared distance to the minimizer of $f$, and the $x$-axis denotes the iterations.

which suggests that each $f_i$ is convex and smooth. We can easily compute that in this case, we have

$$\text{prox}_{\gamma f_i}(x) = \left(\boldsymbol{A}_i + \frac{1}{\gamma}\boldsymbol{I}_d\right)^{-1}\left(\frac{1}{\gamma}x - b_i\right).$$

All experiment codes were implemented in Python 3.11 using the NumPy and SciPy libraries. The computations were performed on a system powered by an AMD Ryzen 9 5900HX processor with Radeon Graphics, featuring 8 cores and 16 threads, running at 3.3 GHz. Code availability: `https://anonymous.4open.science/r/Inexact-FedExProx-code-E783/`

### H.1 COMPARISON OF FEDPROX, FEDEXPROX, FEDEXPROX WITH ABSOLUTE APPROXIMATION AND RELATIVE APPROXIMATION

In this section, we compare the convergence of FedProx, FedExProx and FedExProx with absolute approximation and relative approximation. For FedProx, we simply set the server extrapolation to be 1 while for FedExProx, we set its extrapolation parameter to be $\frac{1}{\gamma L_\gamma}$. We assume exact proximal evaluation for the above two algorithms. For FedExProx with approximations, we fix $\varepsilon_1$ and $\varepsilon_2$ to be reasonable values, respectively. We then set their extrapolation parameter to be the optimal value under the specific setting. Throughout the experiment, we vary the value of the local step size $\gamma$ to see its effect on all the algorithms. Specifically, we select $\gamma$ from the set $\{1000, 100, 10, 1, 0.1.0.01\}$, and we fix $\varepsilon_1 = 0.001$, $\varepsilon_2 = 0.01$ first, then we set them to $\varepsilon_1 = 1e - 6$, $\varepsilon_2 = 0.001$.

Notably in Figure 2 and Figure 3, in all cases, FedExProx with absolute approximation exhibits the poorest performance and converges only to a neighborhood of the solution. This is expected, since the bias in this case does not go to zero as the algorithm progresses. It is worth mentioning that as the local step size $\gamma$ increases, the size of the neighborhood decreases, which supports our claim in Theorem 1. As anticipated, in all cases, FedExProx outperforms FedProx due to server extrapolation. However, as $\gamma$ increases, the performance gap between them diminishes. The performance of FedExProx with relative approximation is surprisingly good, outperforming FedProx in several cases. This suggests the effectiveness of server extrapolation even when the proximal evaluations are inexact.

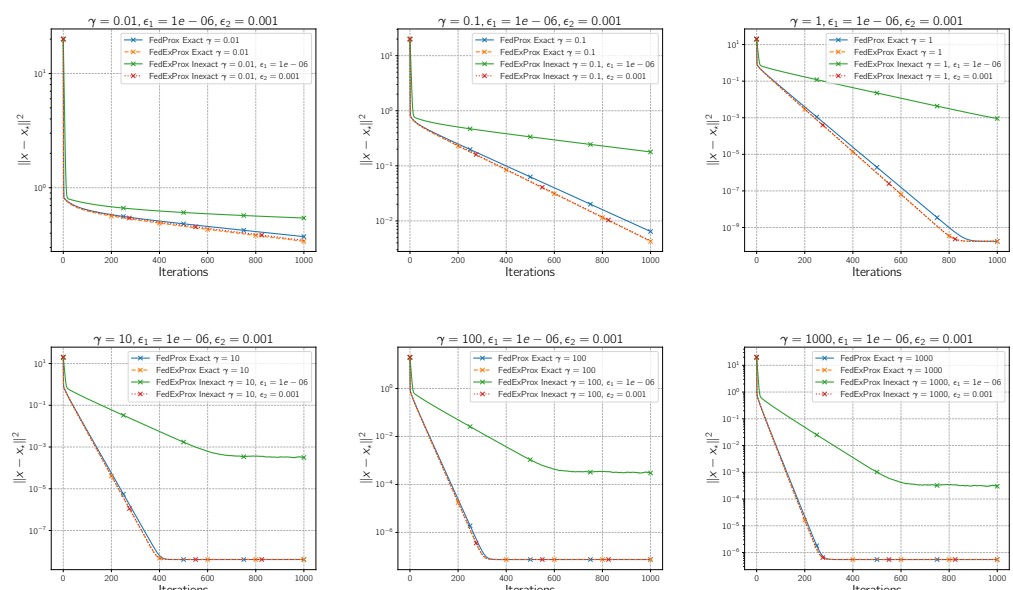

Figure 3: Comparison of FedProx, FedExProx with exact proximal evaluations, FedExProx with $\varepsilon_1$-absolute approximation and FedExProx with $\varepsilon_2$-relative approximation. In this case, we fix $\varepsilon_1 = 1e-6$, $\varepsilon_2 = 0.001$ and pick the local step size $\gamma \in \{1000, 100, 10, 1, 0.1, 0.01\}$. The $y$-axis is the squared distance to the minimizer of $f$, and the $x$-axis denotes the iterations.

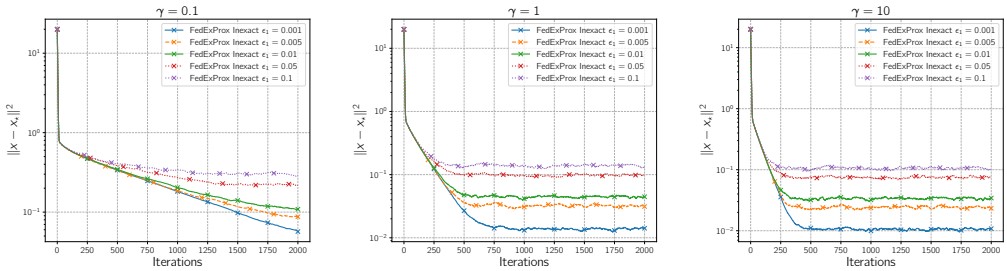

Figure 4: Comparison of FedExProx with $\varepsilon_1$-absolute approximation under different level of inexactness. We select $\gamma$ from the set $\{0.1, 1, 10\}$ and for each choice of $\gamma$, we select $\varepsilon_1$ from the set $\{0.001, 0.005, 0.01, 0.05, 0.1\}$. The $y$-axis denotes the squared distance to the minimizer and the $x$-axis is the number of iterations.

### H.2 COMPARISON OF FEDEXPROX WITH ABSOLUTE APPROXIMATION UNDER DIFFERENT INACCURACIES

In this section, we compare FedExProx with absolute approximations under different level of inaccuracies. We fix the local step size $\gamma$ to be a reasonable value, and we vary the level of inexactness for the algorithm. Specifically, we select $\gamma$ from the set $\{0.1, 1, 10\}$ and for each choice of $\gamma$, we select $\varepsilon_1$ from the set $\{0.001, 0.005, 0.01, 0.05, 0.1\}$.

As observed in Figure 4, the size of the neighborhood increases with $\varepsilon_1$, further corroborating our theoretical findings in Theorem 1. Before reaching the neighborhood, the convergence rates of FedExProx with different level of inexactness are similar, which is expected.

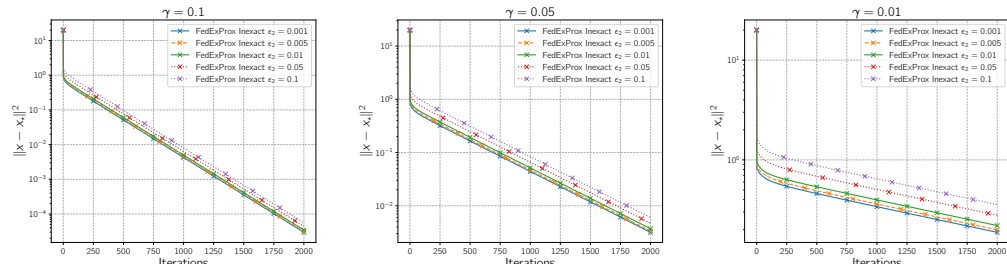

Figure 5: Comparison of FedExProx with $\varepsilon_2$-relative approximation under different level of inexactness. We select $\gamma$ from the set $\{0.01, 0.05, 0.1\}$ and for each choice of $\gamma$, we select $\varepsilon_2$ from the set $\{0.001, 0.005, 0.01, 0.05, 0.1\}$. The $y$-axis denotes the squared distance to the minimizer and the $x$-axis is the number of iterations.

### H.3 COMPARISON OF FEDEXPROX WITH RELATIVE APPROXIMATION UNDER DIFFERENT INACCURACIES

In this section, we compare FedExProx with relative approximations under different level of relative inaccuracies. We fix the local step size $\gamma$ to be a reasonable value, and we vary the level of inexactness for the algorithm. Specifically, we select $\gamma$ from the set $\{0.1, 0.05, 0.01\}$ and for each choice of $\gamma$, we select $\varepsilon_2$ from the set $\{0.001, 0.005, 0.01, 0.05, 0.1\}$.

As observed in Figure 5, in all cases, a smaller $\varepsilon_2$ corresponds to faster convergence of the algorithm. This supports the claim of Theorem 3. All the tested algorithm converges to the exact solution linearly, which validates the effectiveness of the proposed technique of relative approximation to reduce the bias term.

### H.4 ADAPTIVE EXTRAPOLATION FOR INEXACT PROXIMAL EVALUATIONS

In this section, we study the possibility of applying adaptive extrapolation to FedExProx with relative approximations. We do not consider the case of absolute approximation since it converges only to a neighborhood, which causes problems when combined with adaptive step sizes such as gradient diversity and Polyak step size.

We are using the following definition of gradient diversity based extrapolation,

$$\alpha_k = \alpha_{k,G} := \frac{1 + \gamma L_{\max}}{\gamma L_{\max}} \cdot \frac{\frac{1}{n} \sum_{i=1}^{n} \left\| x_k - \text{prox}_{\gamma f_i}(x_k) \right\|^2}{\left\| \frac{1}{n} \sum_{i=1}^{n} \left( x_k - \text{prox}_{\gamma f_i}(x_k) \right) \right\|^2}.$$

for Polyak type extrapolation, we use

$$\alpha_k = \alpha_{k,S} := \frac{\frac{1}{n} \sum_{i=1}^{n} \left( M_{f_i}^{\gamma}(x_k) - \inf M_{f_i}^{\gamma} \right)}{\gamma \left\| \frac{1}{n} \sum_{i=1}^{n} \nabla M_{f_i}^{\gamma}(x_k) \right\|^2}.$$

As it can be observed from Figure 6, in all cases, the use of a gradient diversity based adaptive extrapolation results in faster convergence of the algorithm. This suggests the possibility of developing an adaptive extrapolation for our methods. However, as we can see from Figure 7, a direct implementation of Polyak step size type extrapolation results in divergence of the algorithm, indicating that the challenge may be more complex than anticipated. In our case, this is equivalent to designing adaptive step sizes for SGD with biased updates or CGD with biased compression. To the best of our knowledge, this field remains open and requires further investigation, as biased updates are quite common in practice.

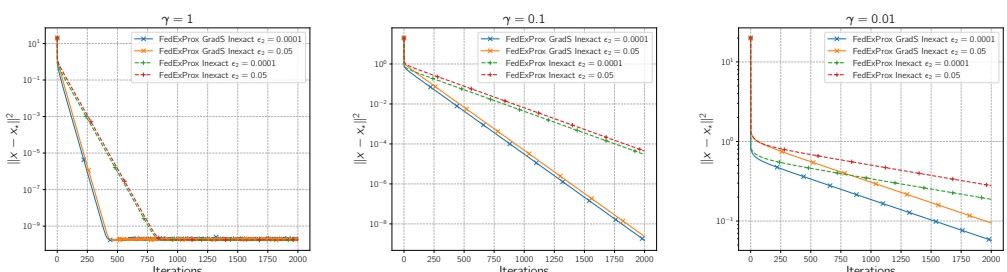

Figure 6: Comparison of FedExProx with $\varepsilon_2$-relative approximation under different level of inexactness using gradient diversity based extrapolation. we select $\gamma$ from the set $\{1, 0.1, 0.01\}$ and for each choice of $\gamma$, we select $\varepsilon_2$ from the set $\{0.0001, 0.05\}$. The $y$-axis denotes the squared distance to the minimizer and the $x$-axis is the number of iterations.

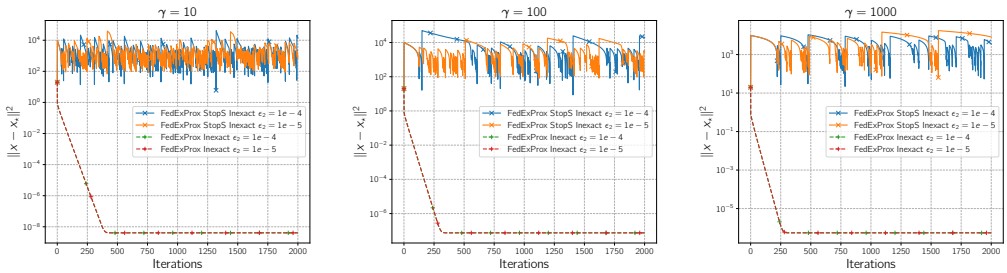

Figure 7: Comparison of FedExProx with $\varepsilon_2$-relative approximation under different level of inexactness using Polyak step size based extrapolation. we select $\gamma$ from the set $\{10, 100, 1000\}$ and for each choice of $\gamma$, we select $\varepsilon_2$ from the set $\{1e-4, 1e-5\}$. The $y$-axis denotes the squared distance to the minimizer and the $x$-axis is the number of iterations.

