# OpenReview forum: "On the Convergence of FedProx with Extrapolation and Inexact Prox"
_ICLR.cc/2025/Conference — Submitted to ICLR 2025_

### Official Review · Reviewer_yKw5 · 2024-10-31

**Soundness:** 2
**Presentation:** 2
**Contribution:** 2
**Rating:** 6
**Confidence:** 3

**Summary:**

The paper considers a finite-sum $\mu$-strongly convex problems for which the interpolation conditions holds, and where each client objective is convex and $L$-smooth.
The work then considers the FedExProx method, which combines proximal client updates with an extrapolated server step, and extends this work to handle inexactness of the client prox computations. Specifically:

- with fixed absolute inexactness they show that the method converges to a neighbourhood of the solution (using a factor $1/4$ smaller extrapolation step).
- with a type of relative inexactness (smaller than order $\mu^2/L^2$) they show exact convergence but for a restrictive extrapolation server stepsize $\alpha$.
- for relative inexactness with a more stringent condition (smaller than order $\mu/L$) they show that the same (large) extrapolation stepsize can be used as in the exact case.
- They provide convergence rate for the local strongly convex and smooth objective with gradient descent and Nesterov acceleration.

**Strengths:**

- The writing is very clear and transparent. They state how results are obtained (relative inexactness by using analysis from biased SGD and from compression) and discuss limitations.
- Considering relative inexactness for federated learning seems interesting

**Weaknesses:**

- the work requires very strong assumptions: the solution needs to be unique (strong convexity) and shared amongst all clients (interpolation condition)
- There is a large body of work on relative inexactness for proximal methods starting with [1], where it is used to essentially inherent the nice properties of an exact proximal computation. Considering the strong assumptions (strong convexity and interpolation condition) it does not seem very surprising that one can extend to a multi-client setting. It would be good to cite this work and put it into context.
- The work does not treat adaptive stepsizes and partial participation as in (exact) FedExProx (they do discuss the difficulty of client sampling in the appendix).

Minor:

- The local convergence rates are not new. It would be good to explicitly state this.
- After Theorem 2 when discussing the slowdown due to small $\alpha$, it would be informative to plug in $\varepsilon_2=c\mu^2/L_{max}^2$ for some $c<1$ and simplify the expression.
- It it possible to to get convergence not only to a neighborhood even for absolute inexactness. It might be worth choosing the $\varepsilon_1$ sufficiently small, to make the comparison with relative inexactness more direct (how does the choice effect the client steps and the communication rounds?).
- For absolute inexactness the server stepsize $\alpha$ is a factor ¼  smaller. Maybe stress that this affects the rate explicitly in Table 1.
- It is maybe worth stating how many iterations (e.g. with Nesterov) are needed to make $\varepsilon_2 =\mu/L$ vs $\varepsilon_2 =\mu^2/L^2$ to make the comparison/tradeoff more explicit between the two relative inexactness results.
- It seems like some concurrent work is treating absolute inexactness which might be worth mentioning [2]

Typos:

- Eq. 4 both f and $\phi$ are present

[1] https://arxiv.org/pdf/2410.15368v1

[2] https://www.emis.de/journals/JCA/vol.6_no.1/j149.pdf

**Questions:**

- Fig. 1(a) indicates that inexactness can help whereas the theory predict otherwise. For inexact proximal gradient inexactness have shown to help for certain regimes (see e.g. page 5 of [3]). Is it possible that something analogue can be said in your setting?
- It is not very clear how much having a more stringent requirement on the relative inexactness ($\mu/L_{max}$ as compared with $\mu^2/L_{max}^2$) buys in terms of the global rate. Is it possible to explicitly compare $S(\varepsilon_2)$ with $(1-4\varepsilon_2 L_{max})$?

[3] https://proceedings.neurips.cc/paper_files/paper/2011/file/8f7d807e1f53eff5f9efbe5cb81090fb-Paper.pdf

---

> ### Author Response · Authors · 2024-11-14
> **Response to Reviewer yKw5**
>
> > We thank you for taking time to review our paper.
>
> ---
> #### Weakness 1:
>
> >The reviewer may have misinterpreted our assumptions. We assume each function $f_i$ is convex (could have multiple minimizers). and operate under the interpolation regime, where the intersection of the set of minimizers of $f_i$ is nonempty. Additionally, we assume the global objective $f$ is strongly convex (Assumption 5), which suggests that the intersection is a singleton.
>
> > The strong convexity assumption simplifies our presentation. Without it, the same reformulation applies to biased SGD in the general convex case, which we can analyze similarly using biased SGD theories.
>
> > Without the interpolation, the algorithm converges to a neighborhood of the true solution. As discussed in Appendix E, the interpolation assumption aligns with FedExProx when viewed through the lens of parallel projection methods.
>
> ---
> ### Weakness 2:
>
> > The referenced work is relevant but was posted after ours, so it wasn’t initially included. Their work provides improved guarantees on PL objectives compared to the original FedExProx algorithm introduced in [2]. Our objective is somewhat 'orthogonal' to theirs, as we focus on removing the impractical assumption of exact proximal operator evaluations. We agree that this reference adds clarity, and it's included in the latest version of our paper.
>
>
> > [2] The Power of Extrapolation in Federated Learning, H. Li, K. Acharya, P. Richtárik
>
> ---
> #### Weakness 3:
>
> > Yes, indeed. For exact proximal operators, FedExProx yields an unbiased SGD, where adaptive step sizes are well-understood. However, for inexact FedExProx, literature on adaptive step sizes for biased SGD is lacking, and we are investigating this gap. Preliminary results (Figures 6 and 7) show that gradient diversity accelerates the algorithm, while the stochastic Polyak step size is less effective, highlighting the need for tailored adaptive step size strategies for biased SGD.
>
> > For the case of client samping, the algorithm performs suboptimally due to the added stochasticity—an expected outcome, as client sub-sampling does not inherently benefit biased compression, as noted in [1]. To address this, one could apply the well-known Error Feedback-21 strategy [1], [2] for biased compression; however, implementing this requires modifying the original FedExProx algorithm, which falls outside the scope of our current focus.
>
> > [1] EF21: A New, Simpler, Theoretically Better, and Practically Faster Error Feedback. P. Richtárik, I. Sokolov, I. Fatkhullin
> > [2] EF21-P and Friends: Improved Theoretical Communication Complexity for Distributed Optimization with Bidirectional Compression. K. Gruntkowska, A. Tyurin, P. Richtarik.
>
> ---
> #### Weakness 4  & Question 2:
>
> > (1): We have added in the lastest version of the paper that the local convergence rates are derived based on existing theories.
>
> > (2): We have changed accordingly in the lastest version of the paper that the local convergence rates are derived based on existing theories.
>
> > (3): Unfortunately, it is not possible to achieve convergence to the exact solution even if $\varepsilon_1$ is sufficiently small; there will always be a neighborhood in this case, determined by the value of $\varepsilon_1$. With relative approximation, however, this neighborhood vanishes as the bias term in (12) diminishes near the optimum. Thus, these two approximation approaches are not directly comparable.
> > Smaller values of $\varepsilon_1$ or $\varepsilon_2$ generally require more local client steps. With absolute approximation, $\varepsilon_1$ determines the neighborhood size but does not affect total communication rounds directly (Theorem 1). In contrast, a smaller $\varepsilon_2$ in relative approximation increases local computation but reduces total communication rounds, akin to the difference between standard SGD and variance-reduced SGD.
>
> > (4): Thank you for the suggestion.  We have now added a note in Table 1 to highlight this.
>
> > (5) & Question $2$: Thanks the suggestion. We include Theorem 2 to illustrate that directly applying results from the biased SGD perspective yields a suboptimal convergence bound and a much restrictive condition on accuracy of the approximation. In contrast, Theorem 3, with a reformulated approach, offers a tighter bound and a relaxed condition, supporting the effectiveness of extrapolation in the inexact case. We include Theorem 2 only to highlight the improvement achieved with Theorem 3.
>
> > (6) Thank you for pointing this out; this is indeed a relevant paper. We have now added this reference to our paper to enhance readability.
>
> > Typos: We have corrected this typo.
>
> ---
> #### Question 1:
>
> > There may be some confusion here. As shown in Fig. 1(a), FedExProx outperforms exact FedProx (without extrapolation) even with inexact proximal updates, though its convergence rate is slower than exact FedExProx, consistent with our theoretical predictions.

---

> > ### Comment · Reviewer_yKw5 · 2024-11-22
> >
> > I thank the author for their response.
> >
> >  > Weakness 1 "...which suggests that the intersection is a singleton."
> >
> >  This is what I meant seemed restrictive. I think its important to explicitly state how to extend to the general convex case.
> >
> > > Weakness 2 "There is a large body of work on relative inexactness for proximal methods..."
> >
> > I accidentally swapped the reference [1] and [2] in the original review. The concurrent work should obviously have been [1] and not [2] which is from the 90s.
> >
> > Considering the large body of work on relative inexactness (starting with [2]), I think it is important to compare. Currently a comparison have been added after absolute approximation (Def. 3), which is misleading. What I would suggest is a discussion regarding _relative inexactness_/approximation (Def. 4) and the fact that it is not new but has been used extensively in the literature before.

---

> > > ### Author Response · Authors · 2024-11-22
> > > **Response to Reviewer yKw5**
> > >
> > > > We thank the reviewer for the clarification.
> > >
> > > > We appreciate the reviewer’s suggestion to highlight the relevant literature on relative approximation. In response, we have added a discussion following Definition 4 to emphasize that this concept is not novel and has been previously studied in the literature.
> > >
> > > > For extension to general convex case, we have added the following discussion at the end of section E.
> > >
> > > > One may consider extend the algorithm into the general convex case. To establish a convergence guarantee, one may notice that in the general convex case, FedExProx still results in biased SGD on the Moreau envelope objective M γ in the general convex and smooth case. The specific approximation used in the algorithm allows for the application of various existing tools for biased SGD. Biased SGD has been extensively studied in recent years; for example, Demidovich et al. (2024) provides a comprehensive overview of its analysis across different settings. Depending on the assumptions, one can adopt different theoretical frameworks to analyze FedExProx, as it is effectively equivalent to biased SGD applied to the envelope objective. For more details on those assumptions, we refer the readers to Demidovich et al. (2024).

---

> > > > ### Comment · Reviewer_yKw5 · 2024-11-25
> > > >
> > > > I thank the authors for their response. My two main concerns still remains:
> > > >
> > > > **Extension to general convex case**
> > > > I am not convinced that the results extends to the general convex case in a meaningful way.
> > > > The papers main concern is with keeping a large extrapolation stepsize, which does not seem to be achieved through the biased SGD analysis (but rather the biased compression analysis).
> > > > In this sense, it still seems that the main result is tied to the strongly convex case.
> > > >
> > > > **Relative inexactness**
> > > > I suggest in the final version to include a more explicit comparison with relative inexactness in the literature. My impression is that the condition is not just "similar" as currently mentioned. I recommend writing relative inexactness in the notation of the paper to make the direct link clear.

---

> > > > > ### Author Response · Authors · 2024-11-25
> > > > >
> > > > > > We thank the reviewer for the response and thoughtful suggestions.
> > > > >
> > > > > > **Extenstion to general convex case**:  Indeed, the tighter convergence analysis in this paper is achieved through the application of biased compression theory. Notably, biased compression is a specific instance of biased SGD, and the state-of-the-art theory on biased compression [1] is, in fact, subsumed by the broader theory of biased SGD [2]. For the particular case of relative approximation, biased compression theory proves to be more effective. However, no existing theory addresses biased compression in the convex setting. This gap arises because biased compression techniques alone tend to perform poorly in certain scenarios, such as the stochastic setting. To address this limitation, combining biased compression with error feedback presents a promising approach for broader applicability. However, this strategy requires modifying the FedExProx algorithm itself. We are aware of this issue and plan to extend the algorithm to ensure its applicability in the convex setting with stochastic sampling.
> > > > > > We have modifed Appendix E accordingly.
> > > > >
> > > > > > **Relative inexactness**: We have revised the description of "similar" and will include a comprehensive discussion comparing various notions of relative approximations that have appeared in the literature. We have also added both absolute and relative approximation in the notation section of the paper. We greatly appreciate the information provided.
> > > > >
> > > > >
> > > > > > [1] On Biased Compression for Distributed Learning. A. Beznosikov, S. Horváth, P. Richtarik and M. Safaryan.
> > > > > >
> > > > > > [2] A Guide Through the Zoo of Biased SGD. Y. Demidovich, G. Malinovsky, I. Sokolov and P. Richtárik.

---

### Official Review · Reviewer_QD48 · 2024-11-01

**Soundness:** 3
**Presentation:** 3
**Contribution:** 3
**Rating:** 6
**Confidence:** 4

**Summary:**

The paper focuses on a federated learning algorithm, called FedExProx, that requires each client to compute exactly a proximal operator. The authors analyze the FedExProx method when the proximal operators of each client is not computed exactly. Theoretical guarantees are provided in the strongly convex and smooth setting and the convergence rate of the algorithm is established. Moreover, the authors highlight a connection with the bias compression methods, that allows them to obtain more refined convergence guarantees. The iteration complexity of the local updates for gradient descent and accelerated gradient descent are also provided. Experimental results validate the theoretical results and showcase the effect of the different notions of inexactness in the computation of the proximal operator in the convergence of the method.

**Strengths:**

- The convergence is established under different notions of inexactness of the proximal operator.
- The paper recovers as a special case the results of the original paper on FedExProx, when the proximal operators are evaluated exactly.
- The connection with the biased compression is interesting.

**Weaknesses:**

- Theorems 1, 2, 3 require the notion of interpolation. Even though an explanation of regimes that satisfy this condition is provided, considering that there are previous works [1], [2] that extend beyond that setting, this assumption seems to be an avenue for future work in this field. More specifically, the initial FedProx algorithm [1] is analyzed in the general non-interpolated setting. In addition, the follow-up work regarding the FedExProx algorithm [2] considers in the main paper the interpolated regime. However, the authors provide additionally an illustration of the algorithm's behaviour in the non-interpolated setting (see Appendix F.3 in [2]). In that sense, it would be useful to provide some additional details on the behaviour of the algorithm in the non-interpolated setting or to comment on the main challenges in extending the current proof technique beyond the interpolation framework, offering in that way a more complete picture and direction for future research.
- Theorems 4, 5 seem to evaluate the inexactness achieved in each client. However, the inexactness is only with respect to the notion of the absolute approximation, for which we know that Theorem 1 is not optimal (since for the same amount of inexactness Theorem 3 gives convergence to the exact solution). Thus, it seems that a characterization of the inexactness in terms of the relative approximation would be also useful. Hence, providing similar theorems for the relative approximation case seems to be a nice addition to the current results.
- Minor: The statement of Theorem 1 can be made shorter in order to increase the readability of the paper.
- Minor typo: In Figure 1, it is mentioned “Figure (c) demonstrates how varying values of $\epsilon_1$ affect FedExProx with relative approximation.” but as shown the varying values correspond to $\epsilon_2$.

References:
[1] Federated Optimization in Heterogeneous Networks, T. Li, A. K. Sahu, M. Zaheer, M. Sanjabi, A. Talwalkar, V. Smith
[2] The Power of Extrapolation in Federated Learning, H. Li, K. Acharya, P. Richtárik

**Questions:**

- Theorem 1 seems to provide convergence guarantees under the natural assumption of absolute approximation. However, the guarantee provided, as mentioned, includes a neighbourhood of convergence which is not optimal. On the other hand, the connection with biased compression provides a refined theorem (Theorem 3), establishing convergence to the exact solution. The amount of inexactness, though, in Theorem 3 is bounded. Do you think that one can achieve the best of both worlds, namely convergence to the exact solution but for arbitrary inexactness.
- How one can compute the relative inexactness $\epsilon_2$ in practice? Are there inherent computational tradeoffs or challenges in the computation of the relative inexactness $\epsilon_2$ in comparison to estimating the constant $\epsilon_1$? It would be nice also if you could comment on ways to approximate $\epsilon_2$ in practical federated learning problems.
- Is it possible to raise the assumption on interpolation in the strongly convex setting by using a more refined proof technique or do you think that extrapolation might be beneficial only on that regime?

**Details Of Ethics Concerns:**

There are no ethics concerns.

---

> ### Author Response · Authors · 2024-11-14
> **Response to Reviewer QD48**
>
> >  We thank you for taking time to review our paper.
>
> ---
> #### Weakness $1$:
> > We agree with the reviewer that detailing the behavior of the algorithm in the non-interpolated setting is essential. To address this, we have added the following discussion to the appendix of our paper in Appendix E:
>
> > In the absence of the interpolation regime assumption, the algorithm converges to a neighborhood of the true minimizer $x_\star$ of $f$. This occurs because $f$ and $M^{\gamma}$ have the same minimizer only under the interpolation regime assumption, as established by Fact 7 and [2]. Since inexact FedExProx can be formulated as biased SGD on the objective $M^{\gamma}$, it converges to the minimizer $x_{\star}^{\prime}$, provided that inexactness is properly bounded. As a result, the algorithm converges to $x_\star^{\prime}$, located within a $\|\|x_\star - x_\star^{\prime}\|\|$-neighborhood of $x_\star$ whose size depends on $\gamma$. Notably, the effects of inexactness and interpolation are, in some sense, 'orthogonal', meaning they do not interfere with each other.
>
> > In addition, FedProx does not require the interpolation regime assumption. However, like FedExProx and its inexact variant, it converges to a neighborhood of the solution. The interpolation assumption was initially introduced based on the motivation behind FedExProx. It is known that the parallel projection method for solving convex feasibility problems is accelerated by extrapolation. Given the similarity between projection operators and proximal operators (which are, in fact, projections onto certain level set of the function), FedExProx was proposed. The interpolation assumption here corresponds to the assumption that the intersection of these convex sets is non-empty in the convex feasibility problem. Although seemingly arbitrary for FedProx, the interpolation assumption aligns naturally with FedExProx when viewed through the lens of parallel projection methods.
>
> ---
> #### Weakness $2$:
> > Theorem 4 and 5 provide both local computational complexities to achiveve absolute approximation (definition 3) and relative approximation (definition 4) using local gradient descent and accelerated gradient descent respectively. It’s possible the reviewer may have missed the second part regarding relative approximation, as both are currently presented in one line due to space constraints. If the paper is accepted and additional space is available, we will separate these results to prevent any misunderstanding.
>
> ---
> #### Weakness $3$:
> > Thank you for your feedback. We have removed some redundancies and streamlined the statement of Theorem 1 for greater clarity.
>
> ---
> #### Weakness $4$:
> > Thanks for pointing this out. We have corrected the typo.
>
> ---
> #### Question $1$:
> > Good question. Unfortunately, we believe it's not possible to achieve an exact solution under conditions of arbitrary inexactness.  For the reformulation of inexact FedExProx, the gradient estimator comprises a gradient term and a bias term. Unlike absolute approximation, relative approximation bounds the bias so it decreases near the solution. Allowing arbitrary inexactness means the bias could vary widely, resulting in convergence only within a neighborhood, similar to absolute approximation.
> >
> > From the perspective of biased SGD, this would be equivalent to assuming that we could use a random vector as a gradient estimator and still converge to the exact minimizer, which is unlikely to hold true.
>
> ---
> #### Question $2$:
> > For relative approximation in Theorem 3, $\varepsilon_2$ must be sufficiently small, specifically $\varepsilon_2 < \frac{\mu}{4L_{\max}}$. If $\mu$ and $L_{\max}$ are known, we select $\varepsilon_2$ accordingly; otherwise, we estimate these values or choose a very small $\varepsilon_2$. A smaller $\varepsilon_2$ increases local computations per communication round but also accelerates progress per round, similar to local training methods.
>
> > In practice, the server can broadcast an accuracy level $\varepsilon_2$ to each client, directing them to perform local SGD, AGD, or other methods during each communication round. The required local iterations will depend on $\varepsilon_2$ and the local objective's characteristics.
>
> ---
> #### Question $3$:
> > In general, we believe that extrapolation would be beneficial across a broader range of conditions. As explained in our response to Weakness 1, if we do not assume the interpolation regime, the FedExProx algorithm still converges to a neighborhood around $x_\star$ with radius $\|\|x_\star - x_\star^{\prime}\|\|$, where $x_\star^{\prime}$ is the minimizer of $M^{\gamma}$. This occurs because, without the interpolation assumption, the minimizers of $f$ and $M^{\gamma}$ do not necessarily coincide. In this setting, the size of the neighborhood depends on both $\gamma$ and $\alpha$, imposing additional constraints if we aim to reach a specific level of accuracy.

---

> > ### Comment · Reviewer_QD48 · 2024-11-14
> >
> > Thank you very much for your response. You have answered all of my questions and concerns.
> > I will keep my current score.

---

> > > ### Author Response · Authors · 2024-11-22
> > > **Response to Reviewer QD48**
> > >
> > > Thank you!

---

### Official Review · Reviewer_htCV · 2024-11-03

**Soundness:** 3
**Presentation:** 3
**Contribution:** 2
**Rating:** 5
**Confidence:** 4

**Summary:**

This work sets to explore a recent algorithm in FL called FEDPROX. This algorithm leans on exact computation of what is called proximal operator.
The paper asks the following natural question: what if we do not fully solve the operator, but rather solve approximately?
In the smooth+strongly-convex case, this paper explore this questions assuming two kinds of approximations $\epsilon_1$ and $\epsilon_2$.

**Strengths:**

- The question is indeed natural and relevant to concurrent FL problems

- The authors cleverly define two kind of approximations and show that one is better then the other, allowing us to converge to the true optimum

- The writing is very clear and easy to follow

- Experiments are illustrative, and in a sense validate the theory

**Weaknesses:**

- While the question is natural and important, the solution is quite straightforward, and does not introduce any novel tools or analysis. Excluding the  $\epsilon_2$ approximation which is nice.

- The paper does not consider stochastic gradients which is the more relevant case in practice, it is important to understand how will the results change in light of this?

**Questions:**

- What is the main challenge and main novelty in your paper?

- Why do you not take stochastic gradients into account? How will the results change assuming this?

---

> ### Author Response · Authors · 2024-11-14
> **Response to Reviewer htCV**
>
> > We thank you for taking time to review our paper.
>
> ---
> #### Weakness $1$ & Question $1$: "While the question is natural and important, the solution is quite straightforward, and does not introduce any novel tools or analysis. Excluding the approximation 2 which is nice." & "What is the main challenge and main novelty in your paper?"
>
> >We respectfully disagree with the reviewer’s assessment that the straightforward analysis in our paper is a limitation. While our analysis does not introduce novel tools, this choice is intentional.
> >Our approach reformulates the inexact FedExProx algorithm in terms of established algorithms, such as SGD (stochastic gradient descent) with a biased gradient estimator and CGD (compressed gradient descent) with biased compression. This reformulation enables us to leverage existing analytical tools effectively. Without those connections, the relatively straightforward analysis presented would not be feasible. In summary, to address this natural and important problem, we opted for a reformulation that allows us to apply existing methodologies, rather than developing entirely new tools.
>
> > The primary challenge in our paper lies in formulating the problem appropriately and establishing optimal complexity bounds for the algorithm. For instance, to reach the conclusion that 'extrapolation aids the convergence of FedExProx, even with inexact proximal operators, provided that inexactness is bounded in a certain manner,' we first needed to recognize an intrinsic connection between FedExProx and biased compression in CGD. This insight allows us to apply existing theoretical frameworks to demonstrate the algorithm’s effectiveness. Without identifying this relationship, reaching such a conclusion would not have been feasible. In addition, it was essential to identify an appropriate way to bound the inexactness, allowing us to eliminate the neighborhood effect. This step was crucial in ensuring that our analysis remains rigorous and aligns with optimal complexity bounds.
>
> > The main novelty of our paper lies in providing an analysis of FedExProx when proximal operators are evaluated inexactly, a scenario that has not been previously studied. Our analysis leads to the new insight that extrapolation remains effective even with inexact proximal operators—a conclusion not previously established. Additionally, as the reviewer noted, we introduce a relative approximation approach that eliminates the neighborhood effect, thereby making the algorithm more practical and applicable in real-world settings.
>
> ---
> #### Weakness $2$ & Question $2$: "The paper does not consider stochastic gradients which is the more relevant case in practice, it is important to understand how will the results change in light of this?" & "Why do you not take stochastic gradients into account? How will the results change assuming this?"
>
> > We do consider stochastic gradients in our analysis. Specifically, in Appendix F, we provide a convergence guarantee for inexact FedExProx with $\tau$-nice sampling of clients (which results in stochastic gradients) and $\varepsilon_2$ relative approximation, as detailed in Theorem 8 based on biased SGD theory. In the specific case of client sub-sampling, the algorithm performs suboptimally due to the added stochasticity—an expected outcome, as client sub-sampling does not inherently benefit biased compression, as noted in [1]. To address this, one could apply the well-known Error Feedback-21 strategy [1], [2] for biased compression; however, implementing this would require modifications to the original FedExProx algorithm, which falls outside the scope of our current discussion. For inexact FedExProx with $\varepsilon_1$ direct approximation, a convergence guarantee can be similarly derived, as outlined in Appendix F.2.
>
> > [1] EF21: A New, Simpler, Theoretically Better, and Practically Faster Error Feedback. P. Richtárik, I. Sokolov, I. Fatkhullin
> > [2] EF21-P and Friends: Improved Theoretical Communication Complexity for Distributed Optimization with Bidirectional Compression. K. Gruntkowska, A. Tyurin, P. Richtarik.

---

### Official Review · Reviewer_VGz2 · 2024-11-03

**Soundness:** 2
**Presentation:** 2
**Contribution:** 2
**Rating:** 6
**Confidence:** 2

**Summary:**

This paper investigates the convergence behavior of FedExProx, a recent extension of the FedProx federated learning algorithm, which includes server-side extrapolation to improve performance in federated settings. A key issue with existing analyses of FedExProx is the assumption that each client can compute the proximal operator exactly, which is unrealistic in practical applications. This paper relaxes this assumption, examining the algorithm’s behavior in cases where the proximal operator is only computed approximately. The authors establish convergence results in smooth, globally strongly convex settings, demonstrating that the algorithm still converges, albeit to a neighborhood around the solution. They also show that careful control can reduce the negative impact of inexact proximal updates and draw connections to biased compression methods. Additionally, they provide an analysis of the local iteration complexity needed for clients to achieve a specific level of inexactness, with empirical validation of their findings through numerical experiments.

**Strengths:**

The paper addresses a significant gap in existing work on FedExProx by relaxing the exact proximal computation assumption. This makes the analysis more applicable to real-world federated learning systems, where inexact computations are the norm due to resource constraints.

**Weaknesses:**

- The theoretical analysis is restricted to globally strongly convex problems, which may limit its applicability to a broader range of federated learning applications that involve non-convex objectives. Extending this analysis to non-convex cases would significantly increase the paper’s impact.

- The assumption of smoothness might not always hold in federated learning, particularly when clients have heterogeneous data distributions. A discussion on how the proposed approach might generalize or be adapted for non-smooth settings would strengthen the paper.

- The experimental part is the weakest part of this work...

**The authors have addressed my concerns in rebuttals. I raise my grade to 6**

**Questions:**

See above

---

> ### Author Response · Authors · 2024-11-14
> **Response to Reviewer VGz2**
>
> > We thank you for taking time to review our paper.
>
> ---
> #### Weakness $1$: "The theoretical analysis is restricted to globally strongly convex problems, ... Extending this analysis to non-convex cases would significantly increase the paper’s impact."
>
> > We agree with the reviewer that extending the analysis to non-convex cases could significantly enhance the paper's impact. However, it remains unclear whether extrapolation would indeed accelerate the convergence of the algorithm in this case, even when proximal operators are assumed to be solved exactly. This uncertainty stems from the foundational motivation behind the FedExProx algorithm, which is based on the parallel projection method for solving convex feasibility problems. Initially, it was observed that extrapolation can accelerate the parallel projection method (in this convex interpolated setting). Given the similarity between projection operators and proximal operators (the latter can be viewed as a projection to a level set of the function), the FedExProx algorithm was developed. In this context, extrapolation is considered in conjunction with convexity; whether it remains beneficial in non-convex settings is still unclear. This rationale led us to focus on the convex case first.
>
> > We have added the above discussion in Appendix E to the latest version of the paper as a clarification.
>
>
> ---
> #### Weakness $2$: "The assumption of smoothness might not always hold in federated learning, particularly when clients have heterogeneous data distributions. A discussion on how the proposed approach might generalize or be adapted for non-smooth settings would strengthen the paper."
>
> > The smoothness assumption is pretty common in convex optimization, and we adopt it here for simplicity of discussion and presentation. In fact, even if we do not assume each local objective function $f_i$ to be $L_i$-smooth, the corresponding Moreau envelope $M^{\gamma}_{f_i}$ is still $\frac{1}{\gamma}$ -smooth as illustrated in [1]. Consequently, the inexact FedExProx still yields a form of SGD with a biased gradient estimator on the convex smooth objective $M^{\gamma}$. This allows us to leverage the relevant theoretical framework to analyze the convergence result in this scenario. Although some technical nuances arise, they do not impact the validity of our conclusion.
>
> > We have added the above discussion in Appendix E to the latest version of the paper as a clarification.
>
> > [1] The Power of Extrapolation in Federated Learning, H. Li, K. Acharya, P. Richtárik
>
> ---
>
> #### Weakness $3$: "The experimental part is the weakest part of this work."
>
> > Thank you for your feedback. In this work, our primary focus has been on the theoretical aspects. Could you please indicate which specific parts of the experimental section you feel could be strengthened, or suggest any additional experiments you would find valuable?

---

### Author Response · Authors · 2024-11-14
**Response to all reviewers**

We sincerely thank all the reviewers for taking the time to review our paper.

The reviewers highlighted key strengths, noting that the paper addresses a significant theoretical gap, enhancing its real-world applicability. They found the problem natural and important, the approximation methods well-designed, the writing clear and accessible, and the connection to biased compression interesting.

The reviewers also raised some concerns, for which we have prepared a comprehensive, case-by-case response to each reviewer. We hope this detailed clarification addresses their questions and provides additional insight into our work. We have incorporated these changes into the latest version of the paper.

---

### Meta-Review · Area_Chair_4f2F · 2024-12-19

**Metareview:**

The paper considers a recently introduced algorithm FedExProx addressing federated learning settings, where an extrapolation step at the server is combined with the local proximal updates at the clients. The addressed finite sum optimization problem is assumed to be such that each component function is smooth and convex, the sum is strongly convex, and an "interpolation regime" condition applies, meaning that the global minimizer is also a minimizer of each of the component functions. The paper relaxes the assumption from prior work that the proximal updates at the clients are computed exactly and instead studies two settings: with additive and multiplicative approximation error. For additive error, it proves that the algorithm converges to a neighborhood of the minimum, while for the multiplicative error convergence to the exact minimum is possible (asymptotically).

Although federated learning as a topic is of high interest to the ML community and the paper provides interesting technical contributions, it's scope is somewhat specialized since the problem assumptions are quite strong and the novelty in coming up with the problem and the solution seems somewhat limited. Perhaps adding the results for the convex (but not strongly convex) objectives claimed by the authors would strengthen the paper.

**Additional Comments On Reviewer Discussion:**

The feedback from the reviews and the overall impression of the paper placed it at borderline. The authors engaged in the discussion with the reviewers and some of the scores were increased as a result. However, there overall seemed be a lack of the enthusiasm for the results, considering they appear to be of niche quality.

---

### Decision · Program_Chairs · 2025-01-22

Reject